# The Functional Crosstalk between Myeloid-Derived Suppressor Cells and Regulatory T Cells within the Immunosuppressive Tumor Microenvironment

**DOI:** 10.3390/cancers13020210

**Published:** 2021-01-08

**Authors:** Maximilian Haist, Henner Stege, Stephan Grabbe, Matthias Bros

**Affiliations:** Department of Dermatology, University Medical Center of the Johannes Gutenberg University, 55131 Mainz, Germany; Henner.Stege@unimedizin-mainz.de (H.S.); stephan.grabbe@unimedizin-mainz.de (S.G.); mbros@uni-mainz.de (M.B.)

**Keywords:** myeloid-derived suppressor cells, regulatory T cells, crosstalk, tumor microenvironment, tumor immune evasion, immunotherapy, cell–cell contact, β2 integrins, CD18, CD11

## Abstract

**Simple Summary:**

Immunotherapy improved the therapeutic landscape for patients with advanced cancer diseases. However, many patients do not benefit from immunotherapy. The bidirectional crosstalk between myeloid-derived suppressor cells (MDSC) and regulatory T cells (Treg) contributes to immune evasion, limiting the success of immunotherapy by checkpoint inhibitors. This review aims to outline the current knowledge of the role and the immunosuppressive properties of MDSC and Treg within the tumor microenvironment (TME). Furthermore, we will discuss the importance of the functional crosstalk between MDSC and Treg for immunosuppression, issuing particularly the role of cell adhesion molecules. Lastly, we will depict the impact of this interaction for cancer research and discuss several strategies aimed to target these pathways for tumor therapy.

**Abstract:**

Immune checkpoint inhibitors (ICI) have led to profound and durable tumor regression in some patients with metastatic cancer diseases. However, many patients still do not derive benefit from immunotherapy. Here, the accumulation of immunosuppressive cell populations within the tumor microenvironment (TME), such as myeloid-derived suppressor cells (MDSC), tumor-associated macrophages (TAM), and regulatory T cells (Treg), contributes to the development of immune resistance. MDSC and Treg expand systematically in tumor patients and inhibit T cell activation and T effector cell function. Numerous studies have shown that the immunosuppressive mechanisms exerted by those inhibitory cell populations comprise soluble immunomodulatory mediators and receptor interactions. The latter are also required for the crosstalk of MDSC and Treg, raising questions about the relevance of cell–cell contacts for the establishment of their inhibitory properties. This review aims to outline the current knowledge on the crosstalk between these two cell populations, issuing particularly the potential role of cell adhesion molecules. In this regard, we further discuss the relevance of β2 integrins, which are essential for the differentiation and function of leukocytes as well as for MDSC–Treg interaction. Lastly, we aim to describe the impact of such bidirectional crosstalk for basic and applied cancer research and discuss how the targeting of these pathways might pave the way for future approaches in immunotherapy.

## 1. Introduction

Immunotherapy with immune checkpoint inhibitors (ICI) has emerged as a promising treatment for many different types of cancer [1], since it has demonstrated stable and impressive tumor regressions even at an advanced stage of disease [1]. However, a large number of cancer patients do not derive benefit from ICI therapy, which is presumably due to an intrinsic or acquired resistance [2]. Increasing evidence suggests that an immunologically active TME is an important predictor for the therapeutic responsiveness toward ICI. Here, it has been demonstrated that both tumor-related factors, e.g., a high mutational load of the tumor cells [3,4], the presence of neoantigens [5,6], microsatellite instability [7,8], and host factors, i.e., the frequency, composition, and activation status [9,10] of tumor-infiltrating lymphocytes (TIL), are predictive for the responsiveness toward ICI treatment. Particularly referring to the activation status of TIL, it has been documented that the extent of programmed cell death 1 ligand 1 (PD-L1) expression on tumor cells correlates with objective response rates in melanoma and non-small cell lung cancer [11,12,13]. Hence, PD-L1 expression levels are also applied in the clinical setting to optimize patient stratification prior to the introduction of ICI therapy. However, both the extent of cytotoxic T-lymphocyte (CTL) infiltration into the tumor and PD-L1 expression on tumor cells do not always correlate with clinical benefit [14]. So far, the various immunosuppressive mechanisms, being present both locally within the tumor microenvironment (TME) and in lymphoid organs, have been identified as major factors mediating immune resistance [15]. Next to the immunosuppressive effects conferred by soluble mediators and leukocyte receptor interactions, the extensive infiltration of the tumor by immunosuppressive cell populations, such as tumor-associated macrophages (TAM) [16,17], myeloid-derived suppressor cells (MDSC) [18], and regulatory T cells (Treg) [19,20], has been identified as a major driver of the pro-tumorigenic transformation in the TME. The presence of these immunosuppressive cells hampers effector T cell induction and recruitment as well as the capability of both natural killer (NK) cells and antigen-presenting cells (APC) to exert effective tumor surveillance, consequently leading to a profound inhibition of the anti-tumor immune response [21]. Thus, the understanding of this immunosuppressive network mediating tumor immune evasion, via cell–cell interactions and by the secretion of soluble immunomodulatory mediators, is essential for the development of novel strategies overcoming immune resistance in cancer treatment.

Recent observations in different cancer models suggest a crosstalk between MDSC and Treg, but its character remains incompletely defined [21,22,23]. As the crosstalk between MDSC and Treg has recently been proposed to be a powerful barrier counter-acting anti-tumoral immune responses, this review is dedicated to give insights into the potential role of cell–cell contacts as a prerequisite for the immunosuppressive mechanisms in the TME, leading to tumor immune evasion. This in consequence facilitates cancer progression and the development of metastases [24].

Furthermore, we aim to delineate the relevance of metabolic pathways and soluble mediators for the functional interaction of MDSC and Treg, according to the current state of scientific research. Since β2 integrins are known to be key regulators of cell adhesion and cell signaling, they are essential for immune cell functions [25]. Accordingly, β2 integrins may constitute potential mediators of the crosstalk between MDSC and Treg [26]. Hence, this review additionally aims to outline the potential role of β2 integrins in this critical cell–cell interaction within an immunosuppressive TME.

## 2. The Immunosuppressive TME

The TME is a complex milieu being composed of a heterogeneous assemblage of distinct tumor—and host cell types such as cancer-associated fibroblasts (CAF), endothelial cells, pericytes, and immune cells that constitute the tumor parenchyma and tumor stroma [14,27]. These various cell types exhibit an extensive crosstalk that dynamically regulates the phenotype and function of the individual cells within the TME. This allows the establishment of a chronic pro-inflammatory state that favors the establishment of a tumor-supportive microenvironment [14,28,29,30]. Thus, it is increasingly evident that the crosstalk between cancer cells and cells of the neoplastic stroma in the TME enables tumor cells to evade host immunosurveillance and thereby supports tumor growth, progression, and metastasis [14,31]. Moreover, the regulatory signaling conveyed by soluble mediators and cell–cell interactions is considered essential in controlling the individual cell function and orchestrating the collective activity within the tumor network [14].

### 2.1. Immunomodulatory Mediators Shape the TME

In the context of immunotherapy, the mutual interactions of tumor-infiltrating immune cells have become an increasingly important area of research, as these cells shape the unique properties of the TME [2]. The tumor-infiltrating immune cells include both tumor-promoting as well as tumor-killing subclasses [14]. Here, it has been shown that tumor infiltration by T cells (mainly CTL) and NK cells correlates with overall prognosis and with the response to ICI treatment [32]. However, in the course of tumor development, a chronic inflammatory state is frequently being induced, which includes the elevation of pro-inflammatory mediators, the infiltration of regulatory immune cells, and the recruitment of endothelial cells and fibroblasts [30,33]. The accumulation of both pro-inflammatory mediators, including cytokines (e.g., interleukin (IL-1, IL-6); tumor-necrosis-factor-alpha, (TNF-α)), chemokines (CC-chemokine ligand 2 (CCL2), and C-X-C motif ligand 2; (CXCL-2)), prostaglandines (prostaglandine E2 (PGE2)) and growth factors (e.g., transforming growth factor-β (TGF-β), granulocyte-macrophage colony-stimulating factor (GM-CSF)), orchestrate the crosstalk between the various cells within the TME. In concert with these soluble mediators, cell–cell-based interactions such as the programmed death protein (PD)-1/PD-ligand (L)-1 axis contribute to the intense crosstalk between the immunosuppressive cell populations, subsequently enhancing the tumor-supporting capacity of the TME, which tips the scale toward immunosuppression and tumor angiogenesis [30]. Altogether, these mechanisms antagonize the cancer-directed immune responses and effectively impair the lytic machinery of TIL in the TME [24,34].

### 2.2. Cellular Composition of the TME

Notably, MDSC, TAM, and Treg are the major cellular components of the immunosuppressive TME. It has been demonstrated that the release of pro-inflammatory cytokines within the TME promotes the immunosuppressive potential of regulatory myeloid cells, such as tumor-associated neutrophils (TAN) [35,36,37], TAM [27,33,38,39,40], MDSC [28,41,42] and regulatory dendritic cells (DC) [43,44,45]. Consequently, a strong tumor infiltration by myeloid cells—being the most abundant cell types within the TME [46]—correlates with rapid tumor growth and a poor prognosis [32]. Here, TAM primarily serves to promote tumor growth and progression via the generation of angiogenetic factors such as vascular endothelial growth factor (VEGF), and the secretion of immunomodulatory cytokines (e.g., IL-6, IL8 and IL-10) [38]. These cytokines generated by TAM and tumor cells promote an aberrant activation of myelopoiesis, resulting in a defective differentiation of myeloid progenitor cells toward MDSC, which exert a strong pro-tumor activity [46,47]. In particular, it has been shown that MDSC suppress both CTL and NK cell activity via immunomodulatory mediators, including IL-1, IL-6, reactive oxygen species (ROS), and nitric oxide (NO) [14,48,49,50]. Hence, the proliferation, activation, and retention of highly immunosuppressive MDSC are not only induced by the chronic inflammatory state within the TME, but it further enhances these conditions, thus creating a positive feedback loop [34,51]. In this context, recent studies revealed that MDSC can modulate the de novo induction, development, and activation of Treg, thus further amplifying the immunosuppressive character in the TME [24]. CD4^+^CD25^hi^ Forkhead-Box-Protein P3 (FoxP3)^+^ Treg cells are frequently found in the course of tumor progression and counteract APC activity, T cell activation, and anti-tumor functions of effector T cells (Teff) [24,52]. Therefore, similar to MDSC, clinical reports confirmed a negative correlation between the frequency of Treg, the patient’s individual prognosis, and the response to ICI treatment [24].

Next to their direct immunosuppressive effects, MDSC and Treg implicitly contribute to the establishment of a TME being characterized by hypoxia, the accumulation of lactic acid, and adenosine (ADO). These factors prevent APC maturation, impair Teff functions, and thus counteract the tumoricidal functions of activated immune effector cells [14,24,46,53]. Since MDSC and Treg systemically expand in the course of tumor development and strongly impair T cell driven anti-tumor immune responses, a detailed characterization of the phenotype and immunosuppressive functions of these cells will be provided in the following section.

## 3. Myeloid-Derived Suppressor Cells

Immature myeloid cells, which are generated in the bone marrow (BM) of healthy individuals in response to acute infection, stress, or trauma, regularly differentiate into mature myeloid cells, such as polymorphonuclear neutrophils (PMN) and monocytes, without exerting immunosuppression [49]. In contrast, neoplastic cells, tumor-associated stroma cells, and a frequently observed inflammation within the TME favor the aberrant activation of myelopoiesis that results in the expansion and recruitment of immature myeloid cells to the tumor site [50]. Indeed, a prominent effect of this “tumor-driven macroenvironment” is the accumulation of highly suppressive, immature myeloid cells in the tumor. Owing to their common myeloid origin and suppressive properties, these cells were termed MDSC.

### 3.1. MDSC Subsets and Their Immunophenotypes

MDSC have first been identified in tumor-bearing mice as immature myeloid cells characterized by the co-expression of CD11b and Gr-1, comprising the lineage markers Ly6G and Ly6C [49,54]. Unlike other myeloid cells, MDSC exhibit a larger diversity of phenotypes, which complicates their identification and characterization [24]. This heterogeneity is in part due to the unique inflammatory milieu present within different tumor entities [41]. Further contributing to the high plasticity of MDSC phenotypes are the temporal variations in the context of tumor-immune editing, as the TME is subject to permanent modulations in the course of the malignant transformation [39,55].

There are currently two main subsets of MDSC to be distinguished: granulocytic (G)-MDSC and monocytic (M)-MDSC [56]. G-MDSC represents the predominant subset of MDSC in the majority of tumor patients and tumor animal models (approximately 75%) compared with M-MDSC (approximately 25%) [57,58]. However, G-MDSC are considered to be less suppressive than M-MDSC when evaluated on a per-cell basis [49,58,59]. This observation was confirmed in human studies, demonstrating a tight correlation between M-MDSC numbers and the inhibition of T cell activation [60].

Murine G-MDSC are generally characterized as CD11b^+^, Ly-6G^+^, Ly-6C^low^ (collectively termed as Gr-1^high^), and CD49^−^, whereas murine M-MDSC are defined as CD11b^+^, Ly6G^−^, Ly-6C^high^ (Gr-1^high^), and CD49^+^, while expressing F4/80, CD115, or CCR2 at varying extents [49]. Due to the polymorphonuclear morphology of G-MDSC and the expression of CD11b and Ly6G, their relationship to PMN is an ongoing issue [24,61]. However, as compared to PMN, G-MDSC show a diminished phagocytic activity, produce higher levels of ROS, and suppress T cell activation upon activation (Table 1) [61]. Therefore, the assessment of these distinctive immunosuppressive properties is important for a definite characterization of G-MDSC, since no distinctive G-MDSC marker set has been established yet [24,62].

Leaving alone the vast heterogeneity in murine MDSC phenotypes, the definition of specific markers for MDSC in humans remains another important issue. Human MDSC are commonly found to be CD11b^+^, CD33^+^, and HLA-DR^−^ [63,64], and the majority of G-MDSC is CD15^+^, whereas CD14 expression is predominantly confined to M-MDSC (Table 1) [24,65]. However, MDSC subsets in humans have yet not been defined consistently with respect to surface marker expression [60].

Despite conflicting reports about MDSC surface marker expression, the clinical value of MDSC has readily been demonstrated: Recent reports highlighted that the frequency of MDSC per se may reflect the tumor burden of cancer patients, thus showing a strong correlation between a high MDSC frequency and a poor prognosis [64,66,67]. On the other hand, the tumor itself may influence the composition of the MDSC compartment. In general, G-MDSC have been found to be the main MDSC subset in patients with renal cell carcinoma [68], whereas M-MDSC constitute the dominant immunosuppressive MDSC subpopulation in melanomas or head–neck cancer [60,65]. However, since none of the aforementioned individual markers are unique to a distinct MDSC subset, the definitive identification of human MDSC subsets requires the assessment of their employed suppressive mechanisms [41,55]. 

### 3.2. Myeloid Cell Plasticity within Tumors

Of note, MDSC entering the TME may have the plasticity to interconvert between different phenotypes. In particular, it has been shown that MDSC might convert into TAM, DC, or TAN depending on the conditions present in the TME (Figure 1) [36,49,50]. For example, the culture of tumor-derived MDSC in the absence of tumor-derived factors was repetitively shown to result in the generation of mature macrophages, PMN, and DC [50,69,70,71], whereas the presence of tumor-derived factors or the adoptive transfer of MDSC into tumor-bearing hosts promoted their differentiation into immunosuppressive macrophages [49,69]. Hence, TAN and TAM may constitute differentiated MDSC or represent a pro-tumorigenic subset of mature PMN and macrophages polarized by soluble mediators [24,49,53].

Despite the phenotypical similarities of the various (immunosuppressive) myeloid cell types, recent reports highlighted, that these can be discriminated by transcriptomic and multi-omics approaches: Referring to the granulocytic cell line in particular, Fridlender et al. revealed cell-specific transcriptome signatures of PMN, G-MDSC, and TAN, confirming the existence of three distinct phenotypes [62]. Moreover, G-MDSC have shown a higher immunosuppressive activity, expressed higher levels of CD115 and CD244, and lower levels of CXCR1 than PMN [61,73]. G-MDSC exerted less phagocytic activity, show a smaller chemotactic response, expressed higher levels of Arginase (Arg)-1 and myeloperoxidase (MPO), and showed a higher production of ROS [49,73].

Likewise, M-MDSC, despite their similarity in morphology and phenotype with other monocytic cell populations, are a functionally distinct population. Particularly, they showed a strong expression of inducible NO-synthase (iNOS) and Arg-1, which explains their highly immunosuppressive character [49,74]. In parallel, it has been reported that hypoxia and hypoxia inducible factor 1α (HIF-1α) within the TME might be key drivers for the upregulation of immunosuppressive Arg-1 and iNOS in M-MDSC and may promote the differentiation of CD11b^+^ Ly6C^+^ M-MDSC into immunosuppressive TAM [53,75]. Since the polarization toward a macrophage M2 phenotype is more likely in MDSC at the tumor site compared to spleen-derived MDSC, it remains an issue to clarify whether the origin of MDSC within the TME might determine the modulation of their phenotype [24,53].

Collectively, these findings provide a mechanistic link between different myeloid cells and indicate that MDSC have the plasticity to interconvert between different phenotypes depending on the specific conditions present within the TME (see Figure 1) [24,53]. However, knowledge of the factors that govern the interconversion of the various granulocytic and monocytic (immunosuppressive) cell types is still far from being complete. Therefore, in vivo strategies and multi-omics approaches are vital to elucidate (combinations of) TME-derived factors that may induce the differentiation, expansion, activation, and interconversion of MDSC populations [24,76].

### 3.3. Mechanisms of Tumor-Induced MDSC Accumulation

Evidence suggests that the release of tumor-derived soluble mediators, such as GM-CSF, VEGF, or IL-6 impairs the myeloid compartment and thus contributes to defective myeloid cell maturation. Moreover, it has been proposed that the relative amounts of G-CSF and M-CSF present within the bone marrow may account for the different shares of the aforementioned MDSC subsets [59]. Here, Waight et al. reported that G-CSF facilitates the accumulation of G-MDSC in the TME, subsequently promoting tumor growth. Moreover, tumor-derived CCL2, CCL12, CXCL5, S100A8, and S100A9 promote the recruitment of immature myeloid cells to the tumor stroma, facilitating the enrichment of both MDSC subpopulations within the TME [49,77,78,79]. Tumor-derived TGF-β has also been found to regulate MDSC accumulation and the polarization of other myeloid cell populations, such as tumor-infiltrating PMN toward an immunosuppressive phenotype [80]. Furthermore, soluble factors such as IL-1β, IL-6, and S100A9 [81,82], and T cell-derived cytokines such as IFN-γ, IL-4, IL10, and IL-13 have been reported to promote immunosuppressive MDSC [83].

The regulation of the integrated myeloid cell network via tumor-derived soluble mediators is controlled on multiple levels via the activation of various transcription factors. Here, the Toll-like receptor (TLR) family, namely TLR-4, which is triggered by S100A8 and S100A9 proteins, contributes to myeloid cell development via the downstream induction of nuclear factor-kB (NFκB), thus supporting the mobilization of myeloid cells to sites of inflammation and their inflammation-driven suppressive potency [49,84]. Other suppressive properties of MDSC are controlled by signal transducer and activator of transcription (STAT)-1 and STAT-6, which regulate myeloid cells by inducing iNOS and Arg-1 expression [69,83].

Further, STAT-3 has been identified as a crucial regulator of MDSC expansion that conveys the recruitment of MDSC to the tumor site by upregulating the pro-inflammatory S100A8 and S100A9 proteins [85]. Hence, S100A9 protein has been proposed as a potential marker characterizing human CD14^+^ HLA-DR^−^ M-MDSC. STAT-3 has also been reported to induce the upregulation of NADPH oxidase (Nox) components, thereby adding up to the immunosuppressive features of MDSC, such as ROS production [49,86]. However, an unsolved question remains: How do these molecular markers relate to the suppressive function of MDSC? Hence, the most definitive characterization of MDSC remains their immunosuppressive function, which will be addressed in the following section.

## 4. Immunosuppressive Properties of MDSC

MDSC are considered key regulators of immune responses in many pathophysiological conditions, including anti-tumor immune responses. G-MDSC and M-MDSC apply antigen-specific and antigen-non-specific mechanisms to regulate immune responses and thus inhibit Teff via a plethora of mechanisms [24]: In peripheral lymphoid organs, the MDSC-mediated suppression of CTL usually requires antigen presentation by MDSC and direct MDSC/T cell contact [87,88]. Otherwise, at the tumor site [53,89] and in the periphery [90], MDSC can suppress nearby T cells in an antigen-independent manner. Although none of these mechanisms are exclusively used by either MDSC subpopulation, it has been demonstrated that ROS generation is characteristic for G-MDSC, whereas Arg-1 expression and the generation of NO has primarily been found in M-MDSC [50,58,75].

### 4.1. Depletion of Nutrients

MDSC confer immunosuppression by various mechanisms (Figure 2), such as the depletion of nutrients. This involves the Arg-1-dependent consumption of L-arginine and deprivation of L-cysteine via its consumption and sequestration in MDSC [91], which causes the proliferative arrest of antigen-activated T cells due to the downregulation of the TCR (T cell receptor) complex and a cell cycle arrest in the G0-G1 phase [49,68]. This phenomenon could be reversed by the replenishment of L-arginine in vitro, but more importantly, in vivo studies reported that the depletion of G-MDSC re-established T cell growth, emphasizing their role in cancer immunosuppression [92]. The inhibition of T cell activation is further enhanced via the consumption of L-tryptophan by MDSC-derived indoleamine-2,3-dioxygenase (IDO) and the subsequent accumulation of kynurenines [93]. Additionally, it was shown that the ADO-generating ectoenzymes CD39 and CD73 are upregulated by MDSC upon HIF-1α induction [94]. ADO impedes Teff function via A2A-receptor (A2AR) and promotes TAM and MDSC suppressive functions via A2BR [95,96]. Whereas the depletion of nutrients and oxygen within the TME comprises T cell function [97,98], tumor hypoxia and lactate accumulation drive HIF-1α stabilization in MDSC, thus upregulating PD-L1 expression and promoting a metabolic switch to fatty acid oxidation (FAO). FAO further induces Arg-1 expression, NO, and peroxynitrite generation, resulting in Teff impairment [99].

### 4.2. Oxidative Stress

Another suppressive mechanisms is the generation of oxidative stress via ROS and reactive nitrogen species [49]. The production of ROS is mediated by Nox-2. Here, studies conducted by Corzo et al. found an upregulation of ROS in G-MDSC isolated from seven different murine tumor models and in tumor-derived G-MDSC obtained from patients with head neck cancer [86]. Interestingly, in the absence of ROS production, G-MDSC did not only lose their ability to confer T cell hyporesponsiveness in vivo, they also differentiated into mature DC [86]. On the other hand, MDSC themselves are protected from the cytotoxic ROS effects by induction of the antioxidant nuclear factor erythroid related factor 2 (Nrf2) and the accumulation of the ROS scavenger phosphoenolpyruvate (PEP) [100]. Peroxynitrite is produced by the cooperative activities of Nox-2, Arg-1, and iNOS [24,101]. Peroxynitrites can cause the nitration of several proteins in tumor and immune cells including the TCR, leading to subsequent TCR desensitization and T cell apoptosis [49]. Moreover, nitration mediates several molecular blocks in T cells, including conformational changes in the TCR–CD8 complex, which renders CTL unresponsive to antigen-specific stimulation [87]. Furthermore, it was found that peroxynitrite interferes with IL-2 receptor signaling [102] and leads to the nitration of CCL-2 chemokines. Consequently, antigen-specific CTLs do not infiltrate into the tumor but instead remain in the tumor-surrounding stroma [79]. Notably, iNOS-driven NO generation may further induce cyclooxygenase-2 (COX-2) activity, resulting in an enhanced PGE2 production, which serves as a potent inductor of IDO, Arg-1, IL-10, and VEGF secretion by MDSC [98]. Lastly, it has been documented that MDSC counteract the upregulation of CD44 and CD162 by T cells in an NO-dependent manner, thus impairing T cell extravasation and tissue infiltration [103,104].

### 4.3. Receptor-Mediated Inhibition

The interference with lymphocyte trafficking and viability is another immunosuppressive mechanism exerted by MDSC: Here, the expression of membrane-bound ADAM-metallopeptidase domain 17 (ADAM17) on MDSC decreased CD62 ligand (CD62L) expression on CD4^+^ and CD8^+^ T cells, thereby limiting the recirculation into lymph nodes [105]. Furthermore, several checkpoint molecules were shown to be critically involved in MDSC-mediated immunosuppression: Among these, PD-L1 and CTLA-4 are prominent negative regulators of T cell functions [106]. PD-L1 exerts its effects via ligation of PD-1 on T cells, resulting in T cell anergy and apoptosis [104], promoting the induction and function of Treg [107] and thus contributing to tumor immune evasion. Treg express CTLA-4, which mainly interacts with CD80/CD86 as expressed by APC-like DC. This interaction causes an impairment of APC-dependent T cell activation [108], enhances the immunosuppressive properties of Treg, and augments peripheral tolerance [109]. Blocking checkpoint molecules via monoclonal antibodies has in fact proven to restore effective anti-tumor immune responses in many patients with advanced malignancies. This effect has been attributed in part to the blockade of MDSC-mediated immunosuppression of Teff [104].

Youn and coworkers additionally suggested that PD-L2 might add up to MDSC-induced T cell inhibition, since PD-L2-/PD-1 interaction skewed T cells toward T-helper cells type 2 (Th2) [61,110,111].

Notably, more recent observations revealed the pivotal role of additional checkpoint molecules, such as the V domain-containing immunoglobulin suppressor of T-cell activation (VISTA), Galectin-9 (Gal-9), and CD155 for MDSC-mediated immunosuppression [112]. In particular, VISTA has been reported to enhance the inhibition of T cell [113,114] and B cell responses [115] by MDSC, whereas a blockade of VISTA allowed for the restoration of a protective anti-tumor response [116,117]. Next, it has been documented that Gal-9-expression on MDSC induced T cell apoptosis via ligation to the checkpoint protein T cell immunoglobulin and mucin domain-containing protein (TIM)-3 [118]. Gal-9 has been also been reported to promote a suppressive TME by enhancing the degradation of stimulator of interferon genes (STING) [119]. As suggested by Dardalhon et al., the interaction of TIM-3^+^, IFN-γ-secreting T cells with Gal-9^+^ MDSC might add up to both MDSC expansion and immunosuppressive functions [120]. Last, recent observations indicated that CD155 might also be involved in MDSC-mediated T cell inhibition, since it may serve as a ligand for T-cell Ig and ITIM domain (TIGIT), which is found on T and NK cells promoting the immunosuppressive functions of Treg [121,122]. Despite conflicting reports about the role of Fas-(L)igand-Fas signaling for MDSC homeostasis and function [90,123], it is well documented that MDSC are able to induce T cell apoptosis via FasL [124]. Next to the T cell-specific inhibition, MDSC also interfere with NK cell cytotoxicity via receptor-mediated mechanisms, e.g., the interaction of membrane-bound TGF-β with the NK cell receptor NKp30 [49,125,126].

### 4.4. Induction of Protolerogenic APC

Additionally, MDSC promote immunosuppression indirectly by the interaction with other cells of the myeloid cell lineage, such as the inhibition of conventional DC and macrophages. This observation further complicates the understanding of the myeloid cell network within tumors, since myeloid cells engage with each other but also have the plasticity to transdifferentiate between different phenotypes. The interdependency of cells in the myeloid linage can be exemplified by the IL-10 and cell–cell contact-mediated mechanisms by which MDSC decrease macrophage IL-12 production, tipping them toward an M2-like phenotype [127]. This initiates a positive feedback loop, as macrophages themselves promote IL-10 synthesis in MDSC, further enhancing the shift toward an M2-like phenotype [49]. An inflamed TME enhances the infiltration of MDSC into the tumor, promotes TLR-4 signaling, the expression of CD14 on MDSC, and their activation. Thus, inflammation is considered a key driver of MDSC and macrophage crosstalk within the TME [128]. Next to the interaction between MDSC and macrophages, MDSC impair DC function via the production of IL-10, which inhibits IL-12 production in DC and the subsequent DC-mediated activation of T cells [49,129]. Adding up more recently to the wide array of immunosuppressive features, it has been observed that MDSC significantly enhance their immunosuppressive potential via the activation and expansion of Treg populations [49]. The character of this interaction is discussed in the following after a brief presentation of Treg characteristics.

## 5. Regulatory T Cells

It has been shown that regulatory T cells play a crucial role in regulating the homeostasis of the immune system and maintaining tolerance [130]. Moreover, Treg have been found to limit the anti-tumor immune response. In accordance, the number of Treg circulating in the blood of cancer patients and the infiltration of Treg into the tumor have been documented to be closely related to the progression and prognosis of multiple cancer entities [20]. More interestingly, the extent of Treg infiltration into human tumors has been proposed to show an inverse correlation with the response to ICI therapy [131,132]. Not least, this observation emphasizes the importance of Treg in the understanding of the anti-tumor immunity and thus the development of novel therapeutic approaches.

### 5.1. Characteristics and Classification of Treg

Treg are defined as a T helper cell subpopulation characterized by the co-expression of CD4, CD25, and in large parts of FoxP3, which inhibit the activation and differentiation of CD4^+^ and CD8^+^ T cells, subsequently impairing reactivity against autologous and tumor-expressed antigens [130,133,134]. According to their biological properties, Treg are generally divided into two groups: natural (n) regulatory T cells and induced (i) regulatory T cells, which commonly express FoxP3 [135]. Whereas nTreg develop in the thymus and exert their inhibitory activity for maintaining immune tolerance largely through intercellular contact, iTreg are derived from peripheral naïve tumor antigen-specific T cells, which are induced by TME-derived cytokines and other soluble mediators [130]. However, both types of Treg act in a tumor-antigen specific manner [136]. In contrast to Th cells and CTL, which rely largely on glycolysis, glucose transporter (GLUT)-1 expression, and on mammalian target of rapamycin (mTOR) signaling, to sustain their metabolic activity, Treg express low levels of GLUT-1, are negatively regulated by mTOR, and depend largely on oxidative phosphorylation and FAO to sustain their metabolic and suppressive activity [98,137,138].

### 5.2. Immunosuppressive Properties of Treg

Treg use several mechanisms to inhibit the anti-tumor immune activity of Teff, NK cells, and DC, thus driving tumor progression. First, it has been shown that Treg-derived soluble mediators, such as IL-10, TGF-β, and IL-35, suppress antigen presentation by DC, promote T cell exhaustion and CTL dysfunction [139,140]. Next, it has been reported that Treg largely interfere with the cell metabolism both within the TME and in secondary lymphatic organs, inhibiting the proliferation of Teff by the competitive consumption of IL-2 [136]. Additionally, the expression of the ectonucleotidases CD39 and CD73 enables Treg to hydrolyze extracellular adenosine triphosphate (ATP) into adenosine monophosphate (AMP) and subsequently to immunosuppressive ADO, which inhibits Teff via engagement with the A2AR [141]. Moreover, the intercellular transfer of cyclic AMP (cAMP) to Teff via gap junctions is considered another metabolic mechanism of Treg to inhibit an effective anti-tumor immune response [130]. Similar to TAM and MDSC, Treg contribute to Arg-1 mediated arginine depletion within the TME [142]. In contrast to Teff, Treg are largely unaffected by limitations of either glutamine or leucine within the TME [143]. Treg counterbalance the high ROS levels within the TME via antioxidants such as glutathione. In agreement, the removal of this ROS-inactivating mechanism in Treg significantly impaired their inhibitory activity [144]. Lastly, Treg hampered Teff and NK cell function and activity via immunosuppressive receptor interactions and the application of cytotoxic enzymes [145]. In particular, Treg are capable of killing effector cells using granzymes or perforins and orchestrate the quiescence of memory T cells by inhibiting effector programs via checkpoint molecules such as cytotoxic T-lymphocyte-associated protein 4 (CTLA-4) [130,146]. Furthermore, Treg hamper anti-tumor immunity via the interaction of CTLA-4 with the co-stimulatory receptors CD80 and CD86, which are expressed by APC-like DC, resulting in the inhibition of their T cell stimulatory capacity [130,147]. In the course of this interaction, it has been found that Treg might enhance immunosuppression via the upregulation of IDO and Arg-1 on APC, which impaired the induction of Teff and in turn also inhibited mTOR signaling in Treg [148,149].

Recent reports indicate that the interaction of Treg with MDSC might further contribute to the immunosuppressive activity and potential of Treg, forming a positive feedback loop that facilitates the enforcement of their suppressive activity [130], as described in the following.

## 6. Functional Crosstalk between MDSC and Treg

The interactions of MDSC and Treg in different cancer models have been proposed to play a critical role in shaping the TME (Table 2) [21]. Although a strong influx of MDSC and Treg has been described for many different tumor entities, there is only little evidence yet for a direct mechanistic link between these major immunoregulatory cell populations. Here, different modes of interactions have been proposed, namely those conferred by soluble mediators, metabolic cooperations, or cell–cell contacts (Figure 3) [21]. Furthermore, it has been suggested that MDSC promote both the conversion of naïve CD4^+^ T cells toward iTreg and the expansion of nTreg [150,151,152].

### 6.1. Functional Interactions Based on Soluble Mediators

Soluble mediators in the TME are considered vital for orchestrating the regulatory tumor immune network. It has been shown as early as 2005 that MDSC promote Treg proliferation in vivo in a TGF-β-dependent manner [158]. Subsequent reports further revealed that IFN-γ and IL-10 are required for the production of both TGF-β and IL-10 by MDSC in tumor-bearing mice [125,151,159]. Additionally, it has been found that IFN-γ and IL-10 upregulated ligands for several co-stimulatory molecules on MDSC (e.g., CD86 and PD-L1). In concert with the aforementioned molecules, the production of soluble mediators (IL-10 and TGF-β) may provide signals for the induction of Treg [151]. Therefore, the authors concluded that MDSC mediate Treg development and subsequent immunosuppression within the TME through a combination of pathways dependent on TGF-β and/or IL-10, which may also involve cell–cell contacts. In the same study, the authors observed that Treg induction and other immunosuppressive mechanisms exerted by MDSC (e.g., NO production) are independent pathways, since iNOS-deficient MDSC lost their suppressive activity but not the ability to induce Treg.

Conversely, Lee and coworkers observed in a murine model of colitis that Treg mediated MDSC proliferation and potentiated their immunosuppressive function via the secretion of TGF-β. This interaction established a positive feedback loop, which mutually enhanced the immunosuppressive capacities of both immune cell populations [153]. More specifically, the authors found that an impaired TGF-β secretion by Treg led to a reduced expression of Arg-1, PD-L1, and iNOS in M-MDSC, resulting in a diminished suppressive activity and a reduced ability of MDSC to induce Treg. Additionally, they documented a significantly stronger G-MDSC accumulation in mice with functionally impaired Treg, suggesting that Treg are important for maintaining normal proportions of MDSC subsets [153].

In another report, IL-35, a heterodimer of EBV-induced gene 3 (EBI3) and of IL-12p35, has been identified as an inhibitory cytokine generated by nTreg, which promoted IL-10 secretion and CD39 expression by iTreg, and NO production in MDSC [160,161]. In turn, IL-10 augmented PD-L1 expression by MDSC, thereby enhancing their immunosuppressive capacity. Notably, the combination signals transduced by PD-L1 and CD169 on MDSC were found to be essential for an induction of IL-35-producing nTreg [162]. Thus, it was suggested that IL-35 generation might establish another positive feedback loop between MDSC and Treg, contributing to the suppressive capacities of Treg [160].

Umansky et al. further found that CCL-5-secretion by M-MDSC resulted in a direct CCR5-dependent recruitment of Treg, indicating that chemokines also add up to MDSC–Treg interaction in the TME [163].

### 6.2. Metabolic Crosstalk between MDSC and Treg

The TME is predominantly characterized by hypoxia, ADO accumulation, a decreased pH, and low tryptophan levels [49]. ADO is derived from ATP being released by apoptotic cancer cells and subsequently degraded in the TME by the CD39/CD73 ectonucleotidase axis [164]. MDSC and Treg have been found to express high levels of CD39 and CD73, thereby contributing to the accumulation of ADO [141,164,165]. ADO serves as a potent immunosuppressive molecule, inhibiting effector immune cell populations via different adenosine receptors (A1, A2A, A2B, and A3). Next to its immunosuppressive role for effector immune cells, it has been observed that ADO might also augment the proliferation and immunosuppressive properties of Treg via A2AR. Of note, it has been reported that TGF-β can even further induce the expression of CD39 and CD73 on MDSC, promoting ADO accumulation in the TME [95]. In accordance with these observations, ADO production serves as an additional mechanism promoting MDSC-mediated immunosuppression, since ADO augmented the accumulation of MDSC within tumor lesions and their immunosuppressive activity [165,166]. Here, in vitro experiments have shown that A2B receptor stimulation of bone marrow hematopoietic cells prevents the differentiation of these progenitor cells into mature myeloid cells [165]. In agreement, the blockade of A2B receptor with a selective antagonist reduced the number of tumor-infiltrating MDSC and improved T cell-mediated immune surveillance in a melanoma model [95]. Hence, these reports suggest that the cooperative ATP degradation by MDSC and Treg might promote the positive feedback loop between these two immunosuppressive cell populations.

Next to the pivotal role of ADO, tumor hypoxia might further augment ADO-driven effects on MDSC accumulation and Treg-suppressive activity [49]. In particular, it has been found that an upregulation of CD73 on both tumor-infiltrating MDSC and Treg could be induced by hypoxia-inducible factor (HIF)-1α [49]. Moreover, the upregulation of HIF-1α by myeloid cells within the TME has been shown to induce the expression of the A2B receptor, causing a differentiation arrest of myeloid cells, subsequently promoting the accumulation of MDSC [167]. HIF-1α also enhanced the expression of PD-L1, thereby promoting the suppressive capacities of MDSC and their interaction with Treg [168]. Taken together, hypoxic conditions, which are characteristic for the TME, induced the accumulation of MDSC and Treg at the tumor site, stimulated Treg induction, and produced the capacities of both cell types to effectively inhibit the anti-tumor responses by reinforcing their functional crosstalk [169].

COX-2 mediated PGE2 generation has been suggested as another enhancer of Treg and MDSC immunosuppressive properties. On the one hand, autocrine PGE2 secretion by MDSC resulted in an enhanced IL-10 secretion and IDO expression in MDSC [98]. On the other hand, PGE2 is known to enhance FoxP3 expression in Treg and thus promotes their inhibitory activity [170].

Lastly, it has been found that M-MDSC express high levels of IDO in chronic lymphocytic leukemia (CLL). IDO is known to catabolize the rate-limiting step of the kynurenine pathway, which resulted in lower tryptophan levels and the accumulation of kynurenines within the TME [171]. Both the depletion of L-tryptophan and the accumulation of kynurenines suppressed T cell activation and induced Treg in vitro [93,172]. IDO-overexpressing tumors were further shown to exhibit a more aggressive growth as well as enhanced Treg and MDSC accumulation [173]. These findings are indicative of a link between IDO, Treg, and MDSC. Indeed, the depletion of Treg in mice bearing IDO-producing tumors significantly reduced the number of tumor-infiltrating MDSC and prevented their migration as assessed in vitro. Hence, IDO-induced Treg may play an important role in the recruitment and activation of MDSC [173].

### 6.3. Cell–Cell-Dependent Crosstalk between MDSC and Treg

In addition to soluble mediators mediating MDSC–Treg crosstalk, the interactions of MDSC and Treg have also been proposed to be regulated by direct cell–cell contacts. More recently, Siret and coworkers found that the accumulation of both immunosuppressive cell populations in a pancreatic ductal adenocarcinoma model (PDAC) was associated with a strong expression of CD40, PD-L1, and CD124 by MDSC, whereas Treg expressed CTLA-4, CD103, CCR5, and TGF-β-receptor at high levels [21]. Here, the depletion of MDSC led to a significant reduction of intratumoral Treg, thus confirming, that MDSC have the ability to promote the de novo generation and recruitment of Treg [21,152]. Notably, in the same study, videomicroscopic analyses demonstrated a physical interaction of both cell populations. When using a transwell system to separate CD4^+^ T-cells and MDSC, no induction of Treg was observed, suggesting that the MDSC-mediated induction of Treg indeed required cell–cell interactions [21]. However, the authors could not identify cell surface receptors mediating this interaction [173].

By contrast, Fujimura and coworkers observed an upregulation of PD-L1 on tumor-infiltrating MDSC in a ret-melanoma model and thus proposed that PD-L1/PD-1 interaction might contribute to the immunosuppressive activities of Treg and the inhibition of T cell proliferation [22]. In particular, the authors could show that the depletion of Treg led to the downregulation of the inhibitory receptors PD-L1, CD276, and B7-H4 on MDSC. These findings suggest that the presence of Treg promoted the acquisition of a more immunosuppressive MDSC phenotype characterized by elevated PD-L1 levels, augmented IL-10, and reduced IFN-γ secretion, contributing to tumor growth [154]. However, iNOS expression by MDSC has not been found to be modified by the presence of Treg.

Vice versa, it has been observed that MDSC enhanced the immunosuppressive properties of Treg in a mouse ovarian cancer model through the interaction of CTLA-4 with CD80 on MDSC [156]. Here, the authors observed an upregulation of CD80-expression by MDSC after direct interaction with Treg. Notably, tumor growth has been retarded upon CD80 knockout or antibody-mediated blockade of either CD80 or CTLA-4 [156]. The importance of checkpoint receptors for MDSC–Treg interaction has been further documented in numerous studies analyzing the role of VISTA, TIM-3, TIGIT, and the lymphocyte-activation gene 3 (LAG-3) as negative regulators of T cell function (Figure 3).

For example, Xu and coworkers suggested that VISTA, which is known to either engage in homotypic interactions or with Selectin P ligand (PSGL-1) as expressed by MDSC [114], might mediate the crosstalk between MDSC and Treg, thus enhancing their immunosuppressive capacity [174]. More interestingly, the antibody-mediated blockade of VISTA impaired the induction and suppressive function of Treg and reduced the overall number of MDSC [116,175]. LAG-3 (CD223) is known as a co-inhibitory regulator of T cells, Treg, and DC, which is induced upon activation and allows for high-affinity binding to MHCII on myeloid APC [176]. The interaction of LAG-3 with MHCII subsequently prohibits the binding of the same MHC molecule to the TCR and thus suppresses T cell activation and cytokine secretion, thereby ensuring homeostasis [177]. In this regard, Pinton and coworkers found that MDSC confer immunosuppression upon MHCII/LAG-3 interaction [178], whereas the blockade of LAG-3 increased the number of Teff [179]. Interestingly, both MHCII expression on MDSC and LAG-3 expression on T cells have been found to be upregulated upon MDSC-T cell interaction [180]. As LAG-3 is essential for maximal Treg suppressive function, including the secretion of the immunosuppressive cytokines IL-10 and TGF-ß [181], the induction of Treg [182], and their differentiation toward a regulatory phenotype [183], it is conceivable that LAG-3/MHCII interaction between Treg and MDSC could mutually enhance their immunosuppressive activity. Notably, a strong cooperative effect between LAG-3, PD-1, and CTLA-4 has been elucidated in recent reports, highlighting the relevance of the interplay between these checkpoint molecules in the regulation of tumor immunity [184,185].

TIM-3, another important checkpoint molecule regulating T cell homeostasis, has also been found to be critically involved in MDSC–Treg interaction. In particular, it has been documented that TIM-3^+^ Treg conferred stronger immunosuppressive capacities via increased IL-10 production and the inhibition of CTL as compared to their TIM-3^−^-counterparts [186,187]. Moreover, Dardalhon et al. suggested that the interaction of MDSC-expressed Gal-9 and TIM-3 on Treg might drive MDSC expansion and suppressive activity [120], whereas a blockade of TIM-3 restored anti-tumor immunity by decreasing Treg numbers, their inhibitory capacity, and MDSC-mediated T cell inhibition [188]. More recently, Wu and coworkers reported that the interaction of TIGIT and CD155 on MDSC might equally be involved in Treg–MDSC crosstalk [121,189], as it added up to the immunoregulatory functions of Treg [190] and MDSC [122].

Next to Treg/MDSC interaction based on the checkpoint molecules and their ligands, it has been reported that the interaction of CD40 on MDSC with CD40L expressed by T cells is required to induce T cell tolerance and Treg accumulation [155]. Namely, the authors observed that CD40-deficient MDSC adoptively transferred to melanoma-burdened mice failed to induce Treg in vivo, suggesting that the CD40/CD40L axis might be crucial for MDSC-mediated inhibition of Teff as well as the expansion of Treg [155]. In accordance with previous reports, the authors specifically identified M-MDSC to activate Treg via the CD40/CD40L axis, whereas G-MDSC failed to do so [155]. Interestingly, the antibody-mediated blockade of CD40 could reverse MDSC-mediated immunosuppression and promote the differentiation of MDSC into DC and macrophages [155]. Although the results may be contradictory at first glance, because CD40 is commonly known to induce adaptive immunity [191], the observations could reveal a crucial mechanism mediating MDSC immunosuppression. Moreover, it has been reported that a combination treatment of IL-2 and agonistic CD40 antibodies elicited synergistic anti-tumor immune responses coincident with the depletion of both Treg and MDSC in primary renal cell carcinomas [192]. This effect has been attributed in part to Fas–FasL mediated apoptosis [104], which is implicated in the regulation of both MDSC and Treg turnover. As for the strong interconnection of MDSC and Treg in the mutual regulation of apoptosis, it is conceivable that FasL–Fas interaction might further be involved in MDSC–Treg interaction, although the exact character of this interaction yet remains undefined.

Altogether, these findings confirm a tight crosstalk between tumor-infiltrating MDSC and Treg, especially within the TME, which is mediated by soluble mediators, metabolic pathways (such as ADO, IDO, and hypoxia) and cell–cell interactions. The aforementioned studies could demonstrate that the blockade of either cell surface receptor may not only reverse the immunosuppressive activity of the targeted cell population but more importantly might even weaken immunosuppression conferred by the interacting cell population. Thus, the targeting of key molecules involved in the establishment of the positive feedback loop might similarly reveal this potentiating character.

## 7. The Role of β2 Integrins for the Immune Regulatory Tumor Network and Tumor Progression

Due to their crucial functions in leukocyte biology, it has been reasoned that β2 integrins might be involved in the immune–cell crosstalk within the immunosuppressive regulatory network. β2 integrins are heterodimeric surface receptors composed of a variable α-(CD11a-CD11d), which determines ligand specificity, and a common abundantly expressed β-subunit (CD18) [193,194,195]. So far, β2 integrins are classified into four different heterodimeric receptors, namely lymphocyte function-associated antigen-1 (LFA-1; CD11a/CD18 engagement), macrophage-1-antigen (Mac-1, also termed complement receptor type 3 (CR-3); CD11b/CD18), CR-4 (CD11c/CD18), and the heterodimer of CD11d/CD18.

### 7.1. β2 Integrins Are Critical for Leukocyte Functions

Being specifically expressed by leukocytes, β2 integrins confer essential functions in mediating adhesion to other cells (LFA-1) and components of the extracellular matrix (ECM), orchestrate the uptake of extracellular material (Mac-1/CR-3) such as complement-opsonized pathogens, and modulate cell signaling (CR-4) [193]. Moreover, β2 integrins are critically involved in the differentiation of immune cells [196], the migration into inflammatory tissues [197], as well as the extent and character of immune responses. β2 integrins interact with various surface receptors, e.g., intercellular adhesion molecules (ICAM1-5), vascular cell adhesion protein (VCAM)-1, platelet endothelial cell adhesion molecule (PECAM-1), receptor for advanced glycation end products (RAGE), and CD40L [198,199]. In particular, β2 integrins are considered critical components for the formation of the immunological synapse between APC and T cells and the intercellular communication of immune cells in general [193,200]. Here, observations indicated that the interaction between LFA-1 on DC and T cell expressed ICAM-1 lowered the threshold required for T cell stimulation [201]. Thus, β2 integrin deficiency resulted in elevated thresholds for TCR activation and subsequently promoted tolerance in vitro and in vivo [202].

### 7.2. β2 Integrins and Treg

However, β2 integrins also regulate the polarization of CD4^+^ T cells: Singh and coworkers found that CD11a^−/−^ and thereby LFA-1-deficient mice presented with decreased frequencies of CD4^+^CD25^+^ Treg, even when stimulated under Treg-promoting conditions, but T cells rather differentiated toward Th17-cells. Further, T cells resembling nTreg according to their phenotype, derived from CD11a^−/−^ mice, conferred a diminished suppressive activity on stimulated naïve T cells [203,204]. Next to CD11a, CD11b might be involved in the regulation of the Treg/Th 17 balance as well [205]. These observations suggest an important role of β2 integrins in Treg differentiation and function [204]. Here, Wang and coworkers demonstrated that the TGF-ß secretion of Treg required the expression of CD18 [206] and that LFA-1 is essential for an effective inhibition of T cell proliferation [207]. In accordance with these findings, the importance of LFA-1, expressed on T cells, for the induction of tolerance and the suppression of inflammation has been documented in various autoimmune diseases, such as experimental autoimmune encephalitis (EAE) [208,209], systemic sclerosis [210,211], rheumatoid arthritis, psoriasis [193,212], or systemic lupus erythematosus [213,214]. Notably, in most of these diseases, expression levels of CD11a on T cells inversely correlated with the severity of the disease [213,215,216]. In order to exert immunosuppressive functions, Treg express high levels of ICAM-1, P-Selectin, and the integrin a4b1 (very late antigen-4; VLA-4) allowing the quick migration to the site of inflammation [217]. Here, β2 integrins may control the homing and migration of Treg during inflammatory conditions, whereas the absence of β2 integrins impairs Treg infiltration into inflamed tissues [218,219]. Given the essential role of β2 integrins in conferring the suppression of effector cell functions in these pathophysiological models, it is plausible that integrins might also contribute to the inhibition of anti-tumor immune responses [220]. Indeed, in the context of tumor immunity, it has been shown that tumor-infiltrating Treg expressed significantly higher levels of Integrin αE (CD103) than peripheral Tregs and that CD103^+^ Treg displayed a more suppressive phenotype [221]. In accordance with these findings, it has been noted that patients suffering from leukocyte adhesion deficiency-1 (LAD1), a hereditary disease characterized by a mutation-dependent loss of CD18 expression—suffered from reoccurring severe infections (attributed to a loss of PMN functions) and renal or intestinal autoimmune disease [222].

### 7.3. β2 Integrins in (Immunomodulatory) Myeloid Cells

The inability of the immune system of LAD1 patients to control infectious diseases mainly results from the functional defects of PMN, monocytes, and macrophages, which constitute the first line of cellular innate immunity [223]. Here, previous studies revealed that CD11b^−/−^ mice were characterized by a strong lung infiltration of PMN in a model of polymicrobial sepsis [224]. However, these mice showed higher bacterial counts and a stronger systemic inflammation, which is indicative of the attenuated killing activity of CD11b-deficient leukocytes [224]. In particular, it has been found that PMN showed a strong functional impairment to kill pathogens in various infection models, such as pulmonary infections with *S. pneumoniae* [225] and *Aspergillus fumigatus* [226], whereas the recruitment and migration into infected lungs was not affected. Moreover, observations from LAD-1 patients suggested that PMN functionality might equally require an integrin-dependent cell–cell contact with other immune cells. Here, it has been found that PMN from LAD-1 patients have indeed not been able to suppress the proliferation of T cells, whereas CD18-expressing PMN could effectively suppress T cell proliferation, while ROS production and degranulation were intact in both PMN populations. Accordingly, the blockade of ICAM-1 reduced T cell suppression by approximately 50%, suggesting that additional molecules might be involved in Mac-1/ICAM interaction [227].

In contrast to the well-established role of β2 integrins on myeloid cell types for T cell interaction and infection control, the role of β2 integrins for MDSC is still rather elusive and has mostly been investigated in the context of tumor development. Observations in various cancer entities have found that the infiltration of CD11b^+^ myeloid cells supports tumor progression and is thus correlated with tumor size, lymph node metastasis, and poor prognosis, which has largely been attributed to the immunosuppressive function of TAM and MDSC [228]. Accordingly, Zhang and coworkers reported that CD11b^−/−^ mice showed a reduced infiltration of myeloid cells in intestinal adenoma and an attenuated tumor growth [229]. Other observations revealed that a systemic application of CD11b blocking antibodies after radiation increased anti-tumor immune responses, which has been explained by a reduced myeloid cell migration to the tumor site and an attenuated support of tumor neovascularization [230]. With regard to the role of β2-integrins for tumor neovascularization, Soloviev and coworkers found that CD11b^−/−^ mice displayed an impaired infiltration of myeloid cells in the tumor tissue, subsequently resulting in an attenuated VEGF secretion and thus attenuated neovascularization [231,232]. This observation is in line with the finding that MDSC produce pro-angiogenic factors and proteases that endorse angiogenesis and metastases of tumors [164] and that β2 integrins are particularly upregulated on MDSC in hypoxic tissues [233].

However, the role of (β2) integrins in regulating the migration of MDSC and the release of their progenitors from the BM is less clear: It has been found that CD11b deficiency impaired MDSC recruitment to intestinal tumors [229]. Moreover, myeloid progenitor cells in the BM express β2 integrins and the integrin VLA-4 [234]. b2 integrins have been found to be involved in the mobilization of myeloid progenitor cells from the BM to the blood and might confer synergistic effects with VLA-4 [235], enabling the release and trafficking of those myeloid progenitors into the vascular microenvironment [236,237,238]. In particular, it has been reported that VLA-4 promotes the homing of CD34^+^ progenitor cells to sites of active tumor neovascularization. Conversely, blocking of VLA-4 impaired the adhesion of myeloid progenitor cells to the tumor endothelia, the infiltration into the tumor, and resulted in a reduced blood vessel density [238,239]. Notably, β2 integrins have been suggested to mediate the IL-8-induced mobilization of myeloid progenitor cells [237], which is indicative for the involvement of MDSC. On the other hand, VLA-4 deficient mice show a strong increase in circulating progenitor cells, suggesting an early release from the BM and the inability of progenitors to infiltrate into tissues [240]. Moreover, Schmid et al. reported that CD11b does not affect myeloid cell recruitment to tumors but rather regulates macrophage polarization [241].

Despite conflicting reports about the exact role of β2 integrins for myeloid cell release from the BM and their ability to migrate or infiltrate into tumor tissue, CD11b has been demonstrated to determine a wide range of MDSC-suppressive functions other than affecting cell recruitment. Hence, it is possible that a cell-specific blockade of β2 integrins might yet show unrecognized effects on tumor immunity [220].

Similar to MDSC, there are divergent reports on the role of β2 integrins for TAM. First, it has been shown that the ligation of β2 integrins in macrophages might impair type I interferon receptor activation, TLR signaling, and induced IL-10 expression, thus enhancing their immunosuppressive capacities [242]. Additionally, the VLA-4 has been reported to be essentially involved in the polarization of macrophages toward an immune-suppressive phenotype via the induction of IL-10, TGF-ß, and Arg-1 [243]. Thus, tumor growth was significantly impaired in mice lacking VLA-4 [243]. In contrast, Schmid et al. demonstrated that a pharmacological activation of CD11b promoted the pro-inflammatory macrophage polarization, which in turn impaired tumor growth in murine and human cancer models [241].

### 7.4. Role of β2 Integrins for MDSC/T Cell Interaction

Yet, the role of β2 integrins and their ligands for the interaction of MDSC with other immune cells within the tumor micro- and macroenvironment is not well defined to date [193]. In this respect, it has been found that MDSC interact with CTL via the β2 integrin Mac-1 and the integrin β1 (CD29) [24]. The antibody-mediated blockade of either integrin abrogated ROS production by MDSC and diminished MDSC-mediated suppression of CTL [101], suggesting that (β2) integrins might be involved in MDSC/T cell interaction. In accordance with this study, a previous report noted that MDSC were unable to suppress T cell-proliferation in the absence of physical contact [227]. Furthermore, it has been observed that the antibody-mediated blockade of CD11b prevented MDSC suppressive activity [227]. Similarly, it has been noted that CD18 expression is involved in Treg suppressive function. Here, Wang and coworkers showed that a reduced expression of β2 integrins disrupts the interaction between Treg and DC, which impaired Treg proliferation and TGF-ß production [206].

The trafficking of MDSC and Treg to the tumor site is mediated via VLA-4 and β2 integrins [217,238]. Thus, Foubert and coworkers found that tumors derived from VLA-4-deficient mice had reduced frequencies of MDSC but increased numbers of CD8^+^ T cells and DC [243]. The induction of β2 integrins and their ligands (e.g., ICAM-1) can be enhanced via the ligation of PSGL-1 [193], which is expressed on both MDSC and Treg [244]. Consequently, PSGL-1 might enhance the migration of either cell type into inflamed tissues [245] and also promote immunosuppressive properties via the ligation of VISTA (see Section 4). Notably, both LFA-1 and Mac-1 have been implicated in Treg [207] and MDSC induction [246] and survival.

Moreover, β2 integrins play a pivotal role in the communication of tumor cells and myeloid cells (e.g., MDSC, TAM, and PMN) within the TME [247], which induce tolerance and thus support tumor growth and progression. Although recent reports have focused on other immune cell interactions mediated by β2 integrins, such as the establishment of the immunological synapse between APC and T cells [248], it seems plausible that β2 integrins might also be involved in the crosstalk between MDSC and Treg. However, a more profound understanding of the role of β2 integrins in the TME, especially with regard to their potential function in regulatory immune cells, is still required. As b2 integrins might mediate multiple possible interactions between different immune cells, a cell-type-specific assessment of the role of the different β2 integrins in orchestrating the tumor immune network is required. This might reveal a more specific insight into their pathophysiological role and enable the development of new therapeutic strategies aiming at a cell-type-specific inhibition of the involved molecules.

## 8. Inhibition of the Immune Regulatory Network for Tumor Therapy

The emergence of ICI in cancer immunotherapy has been a remarkable breakthrough in cancer treatment. In particular, immune checkpoint inhibitors targeting PD-1, PD-L1, or CTLA-4 have been found to restore anti-tumor immune responses in some cancer entities, thus leading to profound therapeutic improvements in patients with advanced cancer diseases. This has been attributed in large parts to the blockade of immune checkpoints either on tumor cells (PD-L1) and Teff (PD-1, CTLA-4).

To date, ICI has been approved for the treatment of several advanced malignancies, including malignant melanoma, Merkel cell carcinoma, non-small cell lung cancer, and head–neck cancer [249]. However, a number of patients do not derive benefit from ICI treatment. This discrepancy in the patients’ responses toward ICI is partly explained by immune-suppressive effects, which are elicited by the diverse character of the immune milieu that exists within the TME, since patients with immunologically anergic tumors are likely to be non-responsive to ICI therapy [250]. Most notably, recent reports suggest that MDSC-mediated immunosuppression substantially contributes to tumor immune evasion [28,251].

Although the identity of MDSC is still a subject of controversial discussion, it is well recognized that these immature myeloid cells play a pivotal role in the inhibition of an efficient anti-tumor immune response, the polarization and recruitment of other immunosuppressive cell populations, and thus the regulation of the immunosuppressive tumor network. Despite the common expression of checkpoint molecules such as PD-L1 on MDSC or CTLA-4 on Treg, it has been observed that anti-PD-L1 and anti-CTLA-4 treatments could only restore an efficient anti-tumor immune response in about 10% of metastatic tumor cases entirely, thus leading to a clinical complete response [252,253,254,255,256]. Hence, it has been speculated that the various immunosuppressive mechanisms exerted by MDSC might rather be addressed in a combinational approach and in a more specific way in order to contribute to a realignment of the immune regulatory network.

Therefore, recent strategies aimed to specifically target MDSC, hence improving the therapeutic efficiency of ICI and restoring anti-tumor immunity in cancer patients. So far, four different approaches have been proposed to directly target MDSC in a combination therapy with ICI, namely (i) a reduction of MDSC frequency by low-dose chemotherapy (paclitaxel, cisplatin, or 5-fluorouracil) or the tyrosine kinase and STAT-3 inhibitor Sunitinib, (ii) the blockade of MDSC recruitment via CCR5 and CXCR2 antagonists, and CSF-1R inhibition, (iii) the inhibition of immunosuppression conferred by MDSC via COX-2 inhibitors, phosphodiesterase-5 inhibitors, or A2AR inhibitors and (iv) the promotion of MDSC differentiation to mature antigen-presenting (non-suppressive) macrophages and DC using all-trans retinoic acid (ATRA) [249,250].

It has been reported in various preclinical tumor models that the targeting of MDSC potentiated the effect of ICI and led to a significantly increased survival [249,250]. Notably, monotherapy with ICI or an adjuvant MDSC-targeting drug was not as efficient as a combination of both approaches, emphasizing the synergistic effects of a combination therapy. In particular, the co-application of the histone deacetylase inhibitor entinostat with anti-PD-1 and anti-CLTA-4 checkpoint inhibitors resulted in an inhibition of MDSC activity, an improved infiltration and effector function of CTL, and a strong regression of the tumor in various cancer models [257,258,259]. Similarly, in a murine pancreatic cancer model, targeting CXCR2 in combination with anti-PD1 treatment revealed that the inhibition of MDSC trafficking into the tumor could equally restore intra-tumoral T cell infiltration and improve ICI efficacy in terms of overall survival [260]. Additional immunotherapeutic agents, including drugs that target either checkpoint molecules, such as TIM-3 [261], LAG-3 [176], or VISTA [262] or immune-metabolic checkpoints such as adenosine (A2A-receptor antagonist, CD73 or CD39 inhibitors) and IDO, yielded promising results in preclinical tumor models [263,264,265] and are currently evaluated in conjunction with anti-PD-1/L1 treatments [254].

## 9. Conclusions and Outlook

In this review, we have outlined that the level of MDSC-mediated immunosuppression might not only be determined by the quantitative amount of MDSC infiltration into the tumor and the extent of their immunosuppressive activity, but it might equally involve the quality of their functional crosstalk with other immunosuppressive cells within the TME. This assumption is in accordance with previous reports suggesting that MDSC-mediated immunosuppression needs to be re-evaluated in the context of the functionally closely interconnected network of immune cells within the TME. In particular, a growing body of evidence describes a tight crosstalk between tumor-infiltrating MDSC and Treg within the TME, which is mediated by cell–cell interactions, soluble mediators, and metabolic pathways. This bidirectional crosstalk enhances synergies among both cell types and thereby amplifies the immuno-suppressive effects of the individual cell population. As a result, MDSC and Treg in the TME are inextricably interconnected such that functions of either population are impacted by the other one [24]. This co-dependency benefits the tumor, but it also implies that therapies that target one population may also reduce the immunosuppressive activity of the other cell population (i.e., the application of anti-PD-L1 or anti-CTLA-4 inhibitors in the clinical setting). Therefore, we propose that targeting of the bidirectional crosstalk between MDSC and Treg might tip the scale toward the restoration of an efficient anti-tumor immune response. Most notably, the targeting of cell surface molecules involved in the direct physical interaction of both MDSC and Treg, such as the checkpoint receptors PD-1/PD-L1, LAG-3/MHCII, VISTA/VISTA-L, TIM-3/Gal-9, and CD80/CTLA-4, and receptor pairs, such as CD40/CD40L or Mac-1/ICAM-1, might be promising approaches to enhance the efficacy of immunotherapy.

Moreover, it is conclusive that targeting those cell surface receptors might further be promising, because it seems plausible that the formation of cell–cell interactions might additionally contribute to the efficacy of receptor-independent mechanisms (e.g., paracrine signaling), as they enable a close proximity of immune cells for a limited period of time, thereby improving the directionality of secreted mediators, such as TGF-β, IL-10, or ADO toward the relevant target cell. As for the strong interdependency of cells within the myeloid cell line, it might further be suggested that targeting of the aforementioned receptors on MDSC (e.g., PD-L1) might as well promote the polarization of TAM toward the inflammatory M1 phenotype [49], consequently adding up to the restoration of an effective anti-tumor immunity.

Despite the lack of specific markers that reflect either the phenotype or the functional polarization of MDSC, the application of new multi-omics techniques might prospectively contribute to a more profound understanding of MDSC heterogeneity, their role in tumor progression, and enable the application of selective MDSC-targeting therapies [250]. Therefore, strategies targeting MDSC populations in general and more particularly their crosstalk with Treg, as part of a combination therapy to enhance ICI potency, should be considered as another promising step in the development toward a generation of immunotherapies with improved therapeutic response and outcome.

## Figures and Tables

**Figure 1 cancers-13-00210-f001:**
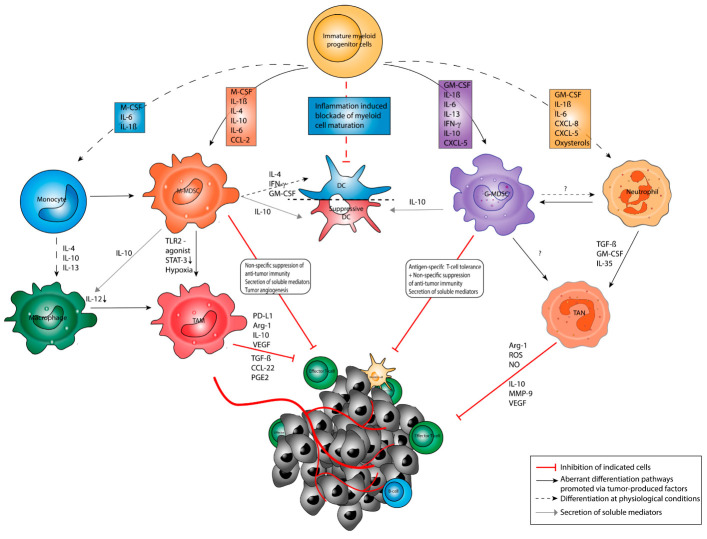
Myeloid cell plasticity in cancer. Myeloid cell types originate from hematopoietic stem cells and multipotent immature myeloid progenitor cells in the bone marrow (BM). The differentiation toward the matured cell line (i.e., polymorphonuclear neutrophils (PMN)) is promoted by soluble mediators and chemokines. In cancer patients, the differentiation pathways are strongly affected by factors produced in the tumor microenvironment (TME) by stromal cells, immune cells, and tumor cells (e.g., granulocyte-macrophage colony-stimulating factor (GM-CSF), interleukin (IL)-1β, IL-6, IL-10, IL-23, interferon-gamma (IFN-γ)). In particular, the TME promotes the polarization of macrophages toward immunosuppressive tumor-associated macrophages (TAM), which confer the inhibition of effector T cells (Teff) within the TME via various mechanisms [72]. PMN within the TME frequently show a polarization toward immunosuppressive TAN, which is driven by soluble factors such as transforming growth factor-β (TGF-β). Tumor-associated neutrophils (TAN) confer immunosuppression via multiple mechanisms. The most prominent effect of the aberrant differentiation includes the accumulation of granulocytic (G-MDSC) and monocytic (M-MDSC) myeloid-derived suppressor cells. Myeloid cells may act as an integrated system in the context of tumor immunity [49]. Depending on the structural composition of the TME, myeloid cells polarize from MDSC toward TAM or TAN or promote the tolerization of DC in the context of a nutrient-depleted, hypoxic, inflamed TME. Under normoxic conditions, IFN-γ and TNF-α have been found to reverse this polarization and promote MDSC differentiation toward immunogenic DC and inflammatory M1 macrophages. It remains questioned if MDSC and TAN undergo an irreversible polarization or can polarize to anti-tumor PMN [24].

**Figure 2 cancers-13-00210-f002:**
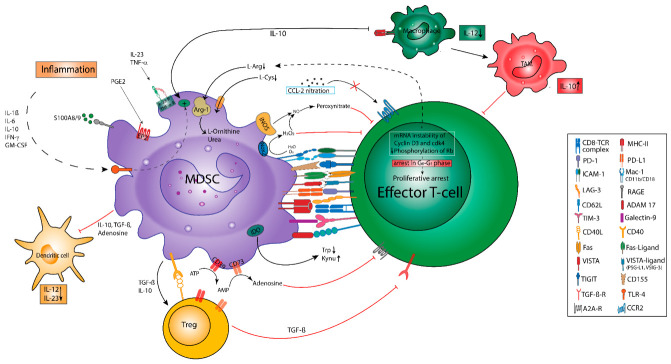
MDSC-mediated inhibition of T cell activation and proliferation. Direct inhibition of Teff involves cell–cell contacts (e.g., via checkpoint molecules), which induce proliferative arrest apoptosis, a reduced migratory activity, and attenuated T cell recirculation. T cell activation is further inhibited via soluble mediators and metabolic pathways: MDSC contribute to L-arginine and L-cysteine depletion in the TME, which causes proliferative arrest via mRNA instability of cyclin-dependent kinase 4 (cdk4), reduced phosphorylation of retinoblastoma protein (Rb), and the loss of the T cell receptor (TCR) ζ-chain on Teff. G-MDSC express high levels of NADPH oxidase (Nox)-2, mediating ROS-dependent inhibition of Teff. The cooperative activities of Nox-2, Arginase (Arg)-1, and inducible nitric oxide synthase (iNOS) generate peroxynitrite, which drives protein nitration resulting in desensitization of the TCR and the interference with IL-2 receptor signaling. The consumption of L-tryptophan and the accumulation of kynurenines in the TME add up to the inhibition of Teff and regulatory T cells (Treg) induction. CD39 and CD73 degrade extracellular ATP to adenosine, which enhances T cell inhibition and Treg induction. Indirect mechanisms of MDSC-mediated immunosuppression include the induction and expansion of Treg both via cell–cell contact-dependent mechanisms and soluble mediators (e.g., TGF-β, IL-10, prostaglandine-E2; PGE2; and A2A-receptor mediated signaling). Likewise, MDSC imprint a tolerogenic function in DC via IL-10, TGF-β, and adenosine. Both the accumulation of Treg and TAM add up to Teff inhibition within the TME. Macrophages are skewed toward an M2 phenotype via IL-10, thus impairing IL-12 production. Tumor-derived soluble factors contribute to STAT3-mediated upregulation of proteins including Nox-2, cell survival proteins (Cyclin D1), or S100A8/9, promoting MDSC accumulation (via S100A8/9 ligation to RAGE), survival, and immunosuppression.

**Figure 3 cancers-13-00210-f003:**
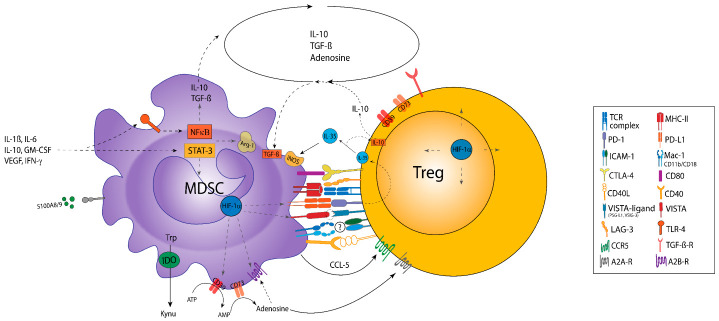
Crosstalk of MDSC and Treg. MDSC and Treg interactions are enhanced by soluble mediators, a close metabolic cooperation, and cell–cell interactions. Particularly, MDSC-derived IL-10 and TGF-β promote Treg induction, proliferation, and activation. The secretion of TGF-β and IL-10 by Treg enhances the generation of these cytokines in MDSC, establishing a positive feedback loop. IL-10 and TGF-β promote the expression of immunosuppressive receptors (e.g., PD-L1) and enzymes (e.g., Arg-1, iNOS, and CD73) on MDSC. Autocrine IL-35 secretion by Treg, which is promoted via the PD-L1-PD-1 pathway, contributes to enhanced IL-10 secretion. The cooperative generation of adenosine (ADO) via the CD39/73 axis and the IDO-mediated accumulation of kynurenines (Kynu) further serve as important mechanisms of the bidirectional crosstalk. First, ADO prevents the maturation of MDSC via A2B-receptor (A2BR) stimulation. A2A-receptor (A2AR) stimulation augments the proliferation and immunosuppressive potential of Treg. Indoleamine-2,3-dioxygenase (IDO)-mediated depletion of tryptophan (Trp) and the Kynu accumulation in the TME add up to the induction of Treg and the recruitment of MDSC to the tumor site. Checkpoint molecules contribute to the crosstalk between MDSC and Treg via PD-L1/PD-1, CD80/CTLA-4, MHC-II/LAG-3, V domain-containing immunoglobulin suppressor of T-cell activation (VISTA)-Ligand/VISTA, Gal-9/TIM-3 (not shown), or CD155/TIGIT (not shown) interaction, promoting the suppressive activities of MDSC and Treg. Notably, CD80 expression is upregulated after direct MDSC–Treg interaction. In addition, CD40–CD40L interaction is involved in MDSC-mediated immunosuppression and Treg expansion at the tumor site. Lastly, the interaction of CD11b/CD18 on MDSC with intercellular adhesion molecule (ICAM)-1 expressed by Treg might enhance MDSC-derived ROS generation. Here, it seems plausible that the engagement of other β2 integrins might also be involved in the crosstalk between MDSC and Treg, e.g., lymphocyte function-associated antigen-1 (LFA-1) on Treg with ICAM-1 on MDSC. The inflammatory and hypoxic TME further enhances MDSC/Treg interaction via mediators, such as IL-1β, IL-6, IL-10, IFN-γ, GM-CSF, or VEGF, which enhance the secretion of IL-10 and TGF-β or promote STAT-3 signaling, contributing to the upregulation of cell surface molecules (e.g., PD-L1, CD80, Mac-1) and enzymes (CD39, Nox-2 or Arg-1) involved in the bidirectional positive feedback loops.

**Table 1 cancers-13-00210-t001:** Phenotypic definitions and functional characteristics of different myeloid cell types present within solid tumors.

Characteristics	PMN	TAN	G-MDSC	M-MDSC	TAM
Murine marker subsets	CD11b^+^	CD11b^+^	CD11b^+^	CD11b^+^	CD11b^+^
	CD11c^−^	Ly6C^low^	Gr-1^+^	Gr-1^+^	F4/80^+^
	Ly6C^low^	Ly6G^+^	Ly6G^+^	Ly6C^high^	CD206^+^
	Ly6G^+^	PD-L1^+^	Ly6C^low^	Ly6G^−^	CD163^+^
	CD101^+^	CD170^high^	CD115^low^	CD49d^+^	CD36^+^
	F4/80^−^		CD49^−^	CD115^high^	MHC-II^low^
	CD115^−^				IL-10R^+^
					CD124^+^
Human marker subsets	CD11b^+^	CD45^+^	CD33^+^	CD33^+^	CD14^+^
	CD66b^+^	CD33^+^	CD11b^+^	CD11b^+^	CD68^+^
	CD15^+^	CD11b^+^	HLA-DR^−^	HLA-DR-	CD205^+^
	CD14^−^	HLA-DR^−^	CD15^+^	CD14^+^	CD163^+^
	CD16^+^	CD66b^+^	STAT-3^high^	STAT-3^high^	CD36^+^
	CD62L^+^	PD-L1^+^	CD66b^+^	CD124^+^	HLA-DR^low^
	CXCR1^+^	CD170^high^	CD244^+^	S100A9^+^	IL-10R^+^
		LOX-1^+^	S100A9^+^		PD-L1^+^
			LOX-1^+^		STAT-3^low^
Maturation stage	predominantly mature	predominantly mature	Immature	Immature	Mature
Potent inductors	GM-CSF	TGF-β	G-CSF IL-6	M-CSF IL-6	IL-4
					IL-10
					TGF-β Hypoxia
Inhibition of T cell proliferation	**Ø**	↑	↑	↑↑	↑
ROS	↑	↑	↑↑	↓	**Ø**
MPO	↑	↑	↑↑	**Ø**	**Ø**
Arginase-1	**Ø**	↑	↑↑	↑	↑
NO	**Ø**	↓	↓	↑↑	↑
NETosis	↑	↑	**Ø**	**Ø**	**Ø**
IL-8	↑	↑	**Ø**	**Ø**	↑↑
Immune cell polarization in response to stimulation	TAN, G-MDSC, APC-like-PMN	PMN	TAN, PMN (?)	TAM, DC	Functional polarization (M1 and M2 phenotype)

Ø: no significant effect; ↓: lower expression/activity compared to the other listed cell types; ↑: higher expression/activity compared to the other cell types; ↑↑: strongest expression/activity among the listed cell types; MPO = myeloperoxidase; NET = neutrophil extracellular traps.

**Table 2 cancers-13-00210-t002:** A selection of important mediators in the functional crosstalk between MDSC and Treg.

Receptors/Soluble Mediators	Cell Type	Species	Disease Model, Immune State	Observations	Reference
TGF-β	Treg and MDSC	mouse	Murine colitis	■ Treg-derived TGF-β enhanced Arg-1, PD-L1, and iNOS expression on MDSC, thus promoting their immunosuppressive properties ■ MDSC themselves showed a stronger induction of Treg after TGF-β stimulation	[153]
PD-1/ PD-L1, IL-10	Treg, MDSC and CD4^+^ T cells	mouse	Ret-melanoma	■ Depletion of Treg downregulated PD-L1 expression on MDSC and inhibited IL-10 production ■ Diminished PD-L1 expression on MDSC led to a reduced inhibition of CD4^+^ T cells ■ iNOS expression was not affected by Treg depletion	[154]
IL-10, TGF-β	MDSC and Treg	mouse	Metastatic colon cancer	■ MDSC mediated Treg induction via IL-10 and TGF-β ■ Treg induction was independent of NO-mediated immunosuppression by MDSC	[151]
Cell-cell contacts (receptors not specified)	MDSC and Treg	mouse	Pancreatic ductal Adeno-Carcinoma	■ Physical interactions between MDSC and Treg (video-microscopic analysis) ■ ^+^MDSC mediated Treg induction and immunosuppression via cell–cell contacts (transwell system)	[21]
CD40/CD40L	MDSC and Treg	mouse	B16-OVA Melanoma	■ CD40-deficient MDSC failed to induce Treg (after adoptive transfer) ■ anti-CD40 antibody treatment promoted the differentiation of MDSC toward DC and macrophages	[155]
CD80/CTLA-4	MDSC and Treg	mouse	Ovarian carcinoma	■ MDSC enhanced the immunosuppressive properties of Treg via the engagement of CTLA-4 with CD80 ■ CD80 depletion led to a significant reduction in tumor growth	[156]
Mac-1	MDSC and T cells	human	Acute systemic inflammation	■ Mac-1 and ROS production were required for the inhibition of T cell function by a suppressive subset of human PMN	[157]

## Data Availability

Not applicable.

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
