# Peer review of "The Functional Crosstalk between Myeloid-Derived Suppressor Cells and Regulatory T Cells within the Immunosuppressive Tumor Microenvironment"

_cancers, 2021, doi:10.3390/cancers13020210_

Round 1

Reviewer 1 Report

In this review article by Haist et al., the authors have focused on the crosstalk between MDSCs and T-regs within the tumor microenvironment.  Unfortunately, the review is to an extent, superficial, does not give us novel insight into mechanisms or what critical mechanisms or signaling events are involved.  In addition, critical components, competing logic or hypotheses are absent in many instances, rendering this superficial, lacking in insight and providing little novel information outside of being exhaustive within some topic areas, including those that may not be critically relevant to the hypothesis at hand. 

  • In the introduction, the statement is made, “In particular, the frequency and composition of TIL, which exert anti-tumor immune responses, is predictive for the responsiveness towards ICI treatment.” While this has been documented, it should be mentioned that expression of PD-L1 is used clinically to justify delivery of some checkpoint inhibitor therapeutics and similarly the presence of neoantigens, micro-satellite instability and general mutational load provide prediction of responsiveness to ICI responses.  Thus, there is more to this than just lymphocytic infiltration.  Indeed, lymphocytic infiltration ignores the difference between cytotoxic effectors as opposed to T-regulatory infiltrating cells. 
  • The bulk of the emphasis in this review is on PD1 and PD-L1 interactions, yet there are a host of additional checkpoint proteins that may be involved and indeed are likely involved, many of which are under investigation at present, both from a translational and clinical vantage point. From a superficial viewpoint, this includes Vista, TIM3, LAG3, PD-L2 wtx  In addition, Fas ligand has been rediscovered as a potentially important mediator in the immunosuppression associated with cell-cell contact.  Thus, there are a host of crosstalk mediators that are not discussed and warrant discussion.
  • The emphasis on integrins is appropriate, particularly for the myeloid cells, and if the authors were to examine the literature over the last 30 or so years regarding expression of integrins and their role in mobilization of progenitor cells, which encompasses MDSCs, they might find a body of literature of interest to their argument. However, this is more likely to be appropriate to arrest and release from the marrow.  This differs from insights into arrest as opposed to crosstalk, and regardless of the mechanism should be encompassed as an argument.
  • The statement is made under 2.2 that “…the secretion of immunomodulatory cytokines, that counteract the anti-tumor cytotoxicity of immune effector cells”. It is unclear how this mechanism may be involved, and certainly the ability of immune augmenting cytokines to counteract antitumor cytotoxicity is an unclear concept.  In general, there may be a differentiation over time as opposed to a direct blockade of anti-tumor cytotoxicity.  This is particularly true if one considers the impact of arginase, ROS, and NOS, by MDSCs and the alternative mechanisms associated with T-regs. 

The above are a few examples of concerns and concepts on initial reading, and the authors may wish to refocus stressing critical aspects and provide novel insight and enlightenment, as opposed to re-discussing previously published concepts.

Author Response

We like to thank the reviewer for the extensive and helpful comments on our manuscript.

Attached please find the Point-to-Point answers to the reviewer´s comments:

  1. In the introduction, the statement is made, “In particular, the frequency and composition of TIL, which exert anti-tumor immune responses, is predictive for the responsiveness towards ICI treatment.” While this has been documented, it should be mentioned that expression of PD-L1 is used clinically to justify delivery of some checkpoint inhibitor therapeutics and similarly the presence of neoantigens, micro-satellite instability and general mutational load provide prediction of responsiveness to ICI responses.  Thus, there is more to this than just lymphocytic infiltration.  Indeed, lymphocytic infiltration ignores the difference between cytotoxic effectors as opposed to T-regulatory infiltrating cells. 

Ad 1: Assessing the expression levels of PD-L1 in solid tumors prior to the application of checkpoint-blockade is a frequently used method in the clinical setting, which allows to predict the patients´ response to checkpoint-blockade. As pointed out by the reviewer, the presence of microsatellite instability (e.g. in colorectal carcinoma) and the presence of neoantigens (e.g. in melanoma) are parameters commonly evaluated prior to the application of systemic therapies. These singular parameters, as well as CTL infiltration in the TME alone, however, might not provide a comprehensive assessment of antitumoral immune activity. The immune infiltrate rather represents a multifaceted component of the TME, which is determined by the interaction of tumor related factors, such as the tumor biology, and host factors, i.e. the ability of the patient ́s immune system to respond effectively to tumors. As suggested by the reviewer, we have revised this paragraph and included more parameters, which might allow a more comprehensive assessment of the immunogenicity of the tumor and its responsiveness to immune-checkpoint-blockade (Section 1.; P2 L44-53)

  1. The bulk of the emphasis in this review is on PD1 and PD-L1 interactions, yet there are a host of additional checkpoint proteins that may be involved and indeed are likely involved, many of which are under investigation at present, both from a translational and clinical vantage point. From a superficial viewpoint, this includes Vista, TIM3, LAG3, PD-L2 wtx  In addition, Fas ligand has been rediscovered as a potentially important mediator in the immunosuppression associated with cell-cell contact.  Thus, there are a host of crosstalk mediators that are not discussed and warrant discussion.

Ad 2: We thank the reviewer for this important hint. Indeed, there are numerous checkpoint molecules on both MDSC and Treg that might be essentially involved in their functional crosstalk and it is indeed likely that upcoming studies might reveal their importance for the bidirectionally enhanced immunosuppressive function of MDSC and Treg. To date, most of the literature has focused and provided evidence for the role of the PD-1/PD-L1 axis in MDSC-Treg interaction. Due to (i) the crucial importance of these negative immunoregulatory molecules (e.g. VISTA, LAG-3, PD-L2, TIM3, TIGIT), (ii) reports which have already assessed their role in MDSC-Treg crosstalk and (iii) the notion, that it is highly conceivable that these might be involved in the receptor-mediated crosstalk, we have provided critical evidence for their role in mediating MDSC and Treg-immunosuppression, which should allow for a broader understanding of immunosuppressive functions mediated by these checkpoint molecules in MDSC and Treg (See sections 4. And 6.3.; P10 L385/386; P11 L405-423; P12 L454; P18 L640-675)

  1. The emphasis on integrins is appropriate, particularly for the myeloid cells, and if the authors were to examine the literature over the last 30 or so years regarding expression of integrins and their role in mobilization of progenitor cells, which encompasses MDSCs, they might find a body of literature of interest to their argument. However, this is more likely to be appropriate to arrest and release from the marrow.  This differs from insights into arrest as opposed to crosstalk, and regardless of the mechanism should be encompassed as an argument.

Ad 3: We agree with the reviewer that beta2 integrins exert multi-facetted roles by contributing to the differentiation/ polarization of leukocyte populations, their migratory capacity, cell-cell-interaction, and signaling processes. The interesting point raised here, partially relate to the importance of beta2 integrins to mediate the migration and infiltration of myeloid cells into solid tumors. Referring particularly to the immature state of myeloid-derived suppressor cells, the role of (beta2) integrins in the mobilization of hematopoietic bone marrow cells should certainly be addressed. Beta2 integrins – and even more important in this context, a4b1 integrins (VLA-4) – are essentially involved in the release and arrest of myeloid progenitor cells in the bone marrow. Previous reports suggest that the role of VLA-4 in the mobilization of myeloid progenitor cells might even exceed the relevance of b2-integrins, even though both molecules might act in a synergistic manner. In order to address this relevant issue, we have included an additional section, which encompasses the role of beta2 integrins in the mobilization and differentiation of progenitor cells from the bone marrow, the migration of MDSC and their infiltration into tumor tissue (see section 7.3.; P22, L838-856).

  1. The statement is made under 2.2 that “…the secretion of immunomodulatory cytokines, that counteract the anti-tumor cytotoxicity of immune effector cells”. It is unclear how this mechanism may be involved, and certainly the ability of immune augmenting cytokines to counteract antitumor cytotoxicity is an unclear concept.  In general, there may be a differentiation over time as opposed to a direct blockade of anti-tumor cytotoxicity.  This is particularly true if one considers the impact of arginase, ROS, and NOS, by MDSCs and the alternative mechanisms associated with T-regs. 

Ad 4: We share the concern of the reviewer that the wording in the original manuscript stating that “TAM primarily serve to promote tumor growth and progression via the generation of angiogenetic factors such as VEGF, and the secretion of immunomodulatory cytokines (e.g. IL-6, IL8 and IL-10), that counteract the anti-tumor cytotoxicity of immune effector cells” was definitely too vague. We have therefore changed the sentence accordingly to avoid ambiguity and provide a logic sequence for the reader (P3, L124-127). The notion that cytokines such as IL-10 and TGF-ß, which among others are secreted by TAM and Treg, act as immunosuppressive modulators of effector T cell function are described in a more sound context of MDSC-mediated immunosuppression (see section 4.4.).

Reviewer 2 Report

The review article summarizes different aspects of the tumor microenvironment from a cellular and cytokine/chemokine aspect. The critical immunosuppressive cells viz. MDSC and Tregs crosstalk is well described with emphasis on b2 integrins. It is a well written and thorough review, however there are sections which can be substantially shortened to retain interest of the readers. For eg. in "The immunosuppressive TME" part, the subsections 2.1 and 2.2 can be merged where the authors can discuss about the cellular composition and their immunomodulatory mediators that then shape the TME. Similarly, section 3.1 MDSC subsets and section 3.2 immunophenotype can be merged as well. There is some redundancy of information that needs attention in order to turn this into a succinct review article. Overall, this is a very informative review. 

Author Response

We like to thank the reviewer for his evaluation and helpful comments.

Attached please find the Point-to-Point-response to the reviewer´s comments:

  1. It is a well written and thorough review, however there are sections which can be substantially shortened to retain interest of the readers. For eg. in "The immunosuppressive TME" part, the subsections 2.1 and 2.2 can be merged where the authors can discuss about the cellular composition and their immunomodulatory mediators that then shape the TME.

Ad 1: We thank the reviewer for the kind assessment of our review and have taken into account the valuable criticism. Thus we have provided a more focused description of the factors contributing to an immunosuppressive TME, thereby shortening both sections for the benefit of the reader (P2 – P4, L79-151; now: P2 and P3 L77-136). However, we might argue in favor of the dichotomization of this chapter (2.1. Immunomodulatory mediators shape the TME and 2.2. Cellular composition of the TME), as this might help the non-expert reader to understand the characteristics of the immunosuppressive TME in a more organized structure.

  1. Similarly, section 3.1 MDSC subsets and section 3.2 immunophenotype can be merged as well. There is some redundancy of information that needs attention in order to turn this into a succinct review article. Overall, this is a very informative review. 

Ad 2: In this subsection, we intended to address the characteristics of MDSC, their role within the TME and mechanisms underlying MDSC accumulation and recruitment to the TME. We agree with the reviewer, that there is some redundancy of information, which we have changed accordingly in our revised manuscript. Similar, we have considered the reviewer´s suggestion and merged both chapters, in order to provide a concise and yet comprehensive description of the different MDSC phenotypes (P4 L200-265; now: P4 L137-181).

Reviewer 3 Report

In this review article, authors tried to summarize the interaction of MDSC with Treg in tumor environment. They considered these cells create the failure of ICI therapy. There are numerous concerns should be clarified:

At least mention one section of ICI therapy, or review the effects of ICI in MDSC section.

Some improper cited references, for example Ref#32, #33 mentioned at Line 109 tries to discuss regulatory DC. However, it is hard to find correlation of these two references for Line 109 sentence.

Due to introduce MDSC first and Treg second in the previous sections, 7.2 and 7.3 can consider change. Also, for emphasize on the discussion fit with title mentioned” tumor microenvironment”, please make less describe of infection, autoimmune and immunodeficiency references in 7.2, 7.3 sections and just keep Line 608~625. And try to amplify 7.4 section with new references.

In the Fig.2 , should modify surface receptor for PGE2/IL-23/TNF-a like receptor not right now look like channel. For intracellular response (nucleus) of effector T cells, please make sure not just the effect of Cyclin D3 mRNA instability affects T arrest, but other pathways.

In the Fig.3, IL-35 act on Treg surface should modify. Line 424~434, 435~442 have to mentioned animal model type. Some other papers related to Ref113 and Ref120, authors put too many words for this two references.

Author Response

We like to thank the reviewer for the extensive comments and valuable criticism.

Attached please find the Point-to-Point responses to the reviewer´s comments:

  1. At least mention one section of ICI therapy, or review the effects of ICI in MDSC section.

Ad 1: As implied in the reviewer statement the pathophysiological mechanisms of immune-checkpoint blockade certainly add to the understanding of both (i) the clinical relevance of this therapeutic approach and (ii) their interference with the immunosuppressive properties of MDSC, Treg, and TAM within the TME. Therefore, we have included a focused presentation of the underlying mechanisms behind checkpoint blockade, in particular with regard to the effects on MDSC (see section 4.; P10 L375-386).

  1. Some improper cited references, for example Ref#32, #33 mentioned at Line 109 tries to discuss regulatory DC. However, it is hard to find correlation of these two references for Line 109 sentence.

Ad 2: We have taken into account the valuable criticism of the reviewer and changed these references, which certainly have been much too vague with regard to the induction and the immunosuppressive properties of regulatory DC. Instead, we have now included three references (one review article and two original research articles), depicting the role of tumor-mediated DC tolerization and their immunosuppressive function within the TME (P3, L121).

  1. Due to introduce MDSC first and Treg second in the previous sections, 7.2 and 7.3 can consider change. Also, for emphasize on the discussion fit with title mentioned” tumor microenvironment”, please make less describe of infection, autoimmune and immunodeficiency references in 7.2, 7.3 sections and just keep Line 608~625. And try to amplify 7.4 section with new references.

Ad 3: The uniformity of the textual structure might indeed add to a more concise description of a certain topic. However, because the role of beta2 integrins has primarily been described in the context of the immunological synapse, immune homeostasis, and T cell immunity, we have reasoned that introducing the role of beta2 integrins first in the Treg-describing section 7.2. might reflect the focus of the research work accordingly. Due to the relevance of beta2 integrins for T cell homeostasis, the majority of research work has been focusing on models in the context of autoimmunity and infection and yet less on tumor models. Despite the discrepancy in structure to the previous chapters 3., 4. and 5., we have reasoned that the disease models presented here, might however reveal essential pathophysiological mechanisms underlying the functioning of T cell mediated and myeloid cell mediated immunity in a comprehensive way. Indeed, it is conceivable, that the pathophysiological mechanisms observed in these autoimmune models also apply in tumor models. However, these assumptions are certainly in parts speculative. Therefore, we have provided a more focused description of these observations made in non-tumor models and condensed the information in a yet comprehensive way (section 7.2. P21 L781-804 and 7.3. P21 L806-820). Moreover, we have provided further evidence for the potential role of beta2 integrins on MDSC and Treg in the regulation of tumor immunity (P22 L838-856 and P23 L884-896).  Owing to the scarcity of evidence for the cell-type specific role of beta2 integrins, particularly in the context of tumor immunity (see section 7.4.), further studies will however certainly be needed to address these highly interesting questions, and such studies are currently prepared in our laboratory.  

  1. In the Fig.2 , should modify surface receptor for PGE2/IL-23/TNF-a like receptor not right now look like channel. For intracellular response (nucleus) of effector T cells, please make sure not just the effect of Cyclin D3 mRNA instability affects T arrest, but other pathways.

Ad 4: Figure 2 provides an overview of the MDSC-mediated inhibition of effector T cells within the TME. Herein, we have also shown potent inductors of MDSC immunosuppressive functions, such as PGE2, IL-23, and TNF-a. These molecules certainly act via membrane-bound receptor molecules (EP1-5 receptors in case of PGE, IL-12/23 receptor in case of IL-23, and TNF-receptor in the case of TNF-a). As suggested by the reviewer we have improved the figure accordingly in order to provide a correct and clear illustration of the biological mechanisms. Moreover, we have added further intracellular pathways involved in G0-G1 cell cycle arrest of effector T cells for a more detailed illustration of the pathophysiological mechanisms behind the arrest of T cell proliferation. Due to the essential role of checkpoint molecules in MDSC-mediated immunosuppression, we have also added further receptor-interactions involved in the inhibition of effector T cell functions for a comprehensive overview of the underlying mechanisms (see section 4.; P10, L387).

  1. In the Fig.3, IL-35 act on Treg surface should modify. Line 424~434, 435~442 have to mentioned animal model type. Some other papers related to Ref113 and Ref120, authors put too many words for this two references.

Ad 5: In Figure 3 we have illustrated an excerpt of mechanisms contributing to the functional crosstalk of MDSC and Treg within the TME and adding up to their immunosuppressive capacities. These involve soluble mediators, metabolic interactions and receptor-mediated crosstalk. Referring particularly to the illustration of IL-35 mediated immunosuppression, we have considered to change this accordingly in order to provide a comprehensible and precise presentation of the underlying mechanism (P19 L676). This should reflect (i) the ability of iTreg to induce iTr35 in an IL-35 and IL-10 dependent manner [Collison, L. et al. IL-35-mediated induction of a potent regulatory T cell population. Nat Immunol 11, 1093–1101 (2010)], (ii) the ability of IL-35 producing iTr35 to promote IL-10 (and CD39) production in Treg [Kochetkova, I. et al. IL-35 stimulation of CD39+ regulatory T cells confers protection against Collagen-II-induced arthritis via the production of IL-10. The Journal of Immunology, 184 (12) 7144-7153, (2010)] and (iii) the direct effects of both IL-10 and IL-35 on MDSC-mediated immunosuppression.

Further, we have considered the valuable criticism that we might have set out the observations from Ref113 and Ref120 in too much detail. We have thus revised lines 424-434 and 435-442 and provide a more focused description of the interesting mechanisms of MDSC-Treg crosstalk based on TGF-ß and IL-35, respectively (P15 L540-562; now: P11/12; L454-468).

Reviewer 4 Report

The manuscript by Haist et al., entitled “The Functional Crosstalk between Myeloid-Derived Suppressor Cells and Regulatory T cells within the immunosuppressive Tumor Microenvironment,” is a well-written review that focuses on the role of the immunosuppressive properties of MDSC and Treg in the TME, and the functional crosstalk between MDSC and Treg for immunosuppression. An additional focus is on the role of cell adhesion molecules, especially b2 integrins, in the differentiation and function of leukocytes, including for MDSC-Treg interaction. This information is presented in 18 pages of text; two informative tables showing the phenotypic definitions and functional characteristics of different myeloid cell types present within solid tumors and indicating important mediators of the functional crosstalk between MDSC and Treg; three excellent figures depicting myeloid cell plasticity in cancer, MDSC-dependent inhibition of T cell activation and proliferation, and crosstalk of MDSC and Treg; and 185 references that annotate the instructive text.

Overall, the manuscript provides an excellent review of the complexities of MDSC and Tregs' suppressive behavior as critical targets from immunotherapy. Two issues could be addressed to enhance the quality of the manuscript. A minor one is a slight discrepancy in the presence of ROS, MPO, Arg1, and NO in Table 1 and the content of these modifiers in their reference #38.

Perhaps more importantly, the authors do not address the roles of MDSC and Treg metabolism. For example, pointing out and discussing the impact of cellular metabolic regulation in suppressive functions, immune homeostasis, and tolerance of MDSC and Tregs would be helpful.

This brings up another point concerning determining the metabolic profiles of s  phenotypes. For example, as instructive as they are, these cells' inclusion with these extensive metabolic phenotypes would likely not provide sufficient numbers of cells for metabolic analysis. It may be helpful and informative to define more selective markers for metabolic analyses. This can be done to prevent overlap among the M- and G-MDSC, TAM, and TAN phenotypes, yet exclude some less important markers. This could appear as a subset of markers that prevent overlap but correctly describe the phenotype’s metabolic profiles.

In summary, this is a well written, informative review. However, the manuscript omits the recently appreciated aspects and importance of metabolic profile in suppressive and homeostatic functions.

Author Response

We like to thank the reviewer for the extensive and very helpful comments.

Attached please find the Point-to-Point responses to the reviewer´s comments:

  1. Overall, the manuscript provides an excellent review of the complexities of MDSC and Tregs' suppressive behavior as critical targets from immunotherapy. Two issues could be addressed to enhance the quality of the manuscript. A minor one is a slight discrepancy in the presence of ROS, MPO, Arg1, and NO in Table 1 and the content of these modifiers in their reference #38.

Ad 1: Table 1 shows the phenotypical and functional characteristics of the different myeloid cell types present within solid tumors and depicts phenotypical markers, which can be used to identify and differentiate each cell population from one another. We have taken into account the valuable criticism of the reviewer and resolved the discrepancy in the illustration of these effector molecules in Table 1 according to reference #38 (see section 3.1.; P5 L258). As for the multitude of effector mechanisms present in all cell types mentioned in the study and the lack of direct comparative analyses in the according studies, this illustration should however rather be interpreted as a comparative presentation facilitating the distinction between these cell types, other than an exact quantitative assessment of the molecular mechanisms. We do however hope, that it adds to a broader understanding of these cell types without being superficial.

  1. Perhaps more importantly, the authors do not address the roles of MDSC and Treg metabolism. For example, pointing out and discussing the impact of cellular metabolic regulation in suppressive functions, immune homeostasis, and tolerance of MDSC and Tregs would be helpful.

Ad 2: The interesting point raised here, has certainly lacked attention in our review. This is, however, partly due to the intended focus on receptor-mediated crosstalk between MDSC and Treg. Immunometabolism is a rapidly emerging field, contributing essentially to the understanding of tumor immunity and mechanisms underlying tumor immune evasion. However, to take into account the emerging role of the cellular metabolism for the immunosuppressive functions of MDSC and Treg, we have addressed this issue in our revised manuscript, discussing, in particular, the relevance of metabolic regulation in the suppressive functions (see section 4.; MDSC P9, L340-346, and L348-368), immune homeostasis (see section 5.1. and 5.2.; Treg P12/13 483-487 and L499-513) and the crosstalk between MDSC and Treg (see 6.2.; P16 L600-603).

  1. This brings up another point concerning determining the metabolic profiles of s  phenotypes. For example, as instructive as they are, these cells' inclusion with these extensive metabolic phenotypes would likely not provide sufficient numbers of cells for metabolic analysis. It may be helpful and informative to define more selective markers for metabolic analyses. This can be done to prevent overlap among the M- and G-MDSC, TAM, and TAN phenotypes, yet exclude some less important markers. This could appear as a subset of markers that prevent overlap but correctly describe the phenotype’s metabolic profiles.

Ad 3: As pointed out by the reviewer, the phenotypical distinction of the different immunosuppressive myeloid cell populations within the TME is a widely discussed topic in cancer research, as TAN, TAM, and MDSC play pivotal roles for the inhibition of effective anti-tumor immune responses and contribute to tumor resistance. However, the phenotypical plasticity of tumor-infiltrating myeloid cells remains a yet unresolved problem for the identification of specific markers being uniquely expressed in each of these cell populations. In many cases, the exact definition of these cell populations therefore still requires the assessment of their phenotypical properties. The emergence of highly multiplexing techniques, which allow for the assessment of functional immune marker expression (such as Lectin-type oxidized LDL receptor 1; LOX-1 for G-MDSC or TAN; IL-4R, FIZZ1 or YM1 for M2 macrophages) and the emergence of immunometabolic signatures might contribute to a better understanding of the distinct phenotypes of these myeloid cell populations. Particularly, referring to specific metabolic phenotypes of myeloid cells in order to differentiate the aforementioned cell populations, the current literature suggests the assessment of glutamine metabolism (being particularly important for PMN, TAN, and TAM), Nrf2- expression (M-MDSC), REDD1 expression (TAM), LOX-1 expression (G-MDSC and TAN), IDO-expression levels (MDSC and TAM), as well as the dominant metabolic pathways upon effector function (which are glycolysis and PPP for PMN; FAO and PPP for both TAN and M-MDSC, and FAO for M-MDSC and TAM). These listed markers and pathways might certainly only be an excerpt of a widely regulated distinct metabolic phenotype and partly be speculative due to the lack of direct comparative analyses between these immunosuppressive myeloid cell populations. However, for the purpose of a broader understanding and the prevention of overlap between these cell types, we have included these insights into each chapter of our review referring to the respective cell population (see sections 4. and 5.). Despite the enormous progress in the field of immunometabolismin the last years, which has added to the understanding of the metabolic regulation within each distinct immune cell population, further studies will be highly interesting – particularly with regard to the direct comparison between myeloid cell populations.